# GUI-360°: A Comprehensive Dataset and Benchmark for Computer-Using Agents

## Abstract

We introduce GUI-360°, a large-scale, comprehensive dataset and benchmark suite designed to advance *computer-using agents* (CUAs). CUAs present unique challenges and is constrained by three persistent gaps: a scarcity of real-world CUA tasks, the lack of automated collection-and-annotation pipelines for multi-modal trajectories, and the absence of a unified benchmark that jointly evaluates GUI grounding, screen parsing, and action prediction.

GUI-360° addresses these gaps with a largely automated pipeline for query sourcing, environment-template construction, task instantiation, batched execution, and LLM-driven quality filtering. The released corpus contains over 1.2M executed action steps across thousands of trajectories in popular Windows office applications, and includes full-resolution screenshots, accessibility metadata when available, instantiated goals, intermediate reasoning traces, and both successful and failed action trajectories. The dataset supports three canonical tasks, GUI grounding, screen parsing, and action prediction, and a hybrid GUI+API action space that reflects modern agent designs. Benchmarking state-of-the-art vision–language models on GUI-360° reveals substantial out-of-the-box shortcomings in grounding and action prediction; supervised fine-tuning yield significant gains.

## 1 Introduction

Recent advances in vision–language and large language models have sparked rapid progress toward intelligent agents that automate tasks inside digital environments Zhang et al. (2024a). Such agents interpret natural-language requests, perceive screen content via pixels and/or accessibility (a11y) metadata, plan sequences of operations, and then either navigate the GUI or invoke APIs to complete tasks on a user's behalf Zhang et al.. They can dramatically reduce user effort for routine productivity tasks and enable novel human–computer workflows. However, realizing this potential requires two tightly coupled capabilities: reliable screen understanding (element grounding or screen parsing) Cheng et al. (2024); Lu et al. (2024); Zheng et al. (2025b) and robust action planning (stepwise action prediction and execution) Zhang et al. (2024b). Both capabilities in turn depend critically on large, diverse, and high-quality datasets grounded in realistic execution contexts Wang et al. (2024c).

We focus on a concrete, under-served class of agents called *computer-using agents* (CUAs) OpenAI (2025a): agents whose primary operating domain is the desktop computer environment. Desktop CUAs differ from web Zheng et al. (2024) or mobile Wang et al. (2024b) agents in several important ways. Desktop applications present very high-resolution mixed-content screens, heterogeneous widgets and document formats, arbitrary window layouts (multi-window and multi-monitor settings), and frequently lack standardized a11y metadata Zhang et al. (2025). Tasks on desktop systems are also often longer-horizon and more compositionally structured. These characteristics make desktop CUAs substantially more challenging than their web or mobile counterparts Zhang et al. (2024a).

Despite growing interest, progress on desktop CUAs is hampered by three persistent gaps. First, there is a scarcity of real-world task collections: existing datasets are often handcrafted or LLM-synthesized Sun et al. (2024), which limits their ability to capture the frequency and diversity of authentic user intents. Second, automated pipelines for data collection and annotation are largely missing Nayak et al. (2025). Manual execution and labeling of desktop interactions is expensive, error-prone, and difficult to scale. Third, no unified, large-scale benchmark exists that supports the breadth of tasks needed for comprehensive evaluation Nayak et al. (2025); Li et al. (2025).

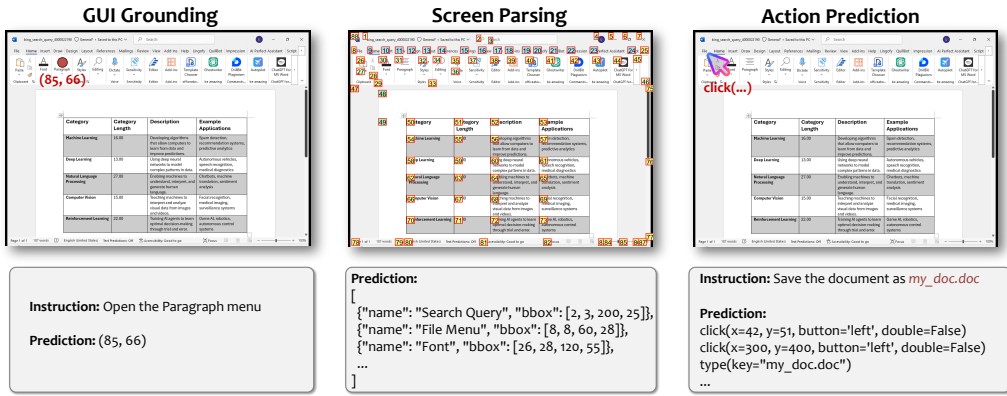

Figure 1: GUI Grounding, Screen Parsing and Action Prediction tasks included in our GUI-360°.

Table 1: Comparison of CUA datasets across dimensions.

| Dataset | Query Source | Task | | | Samples | Data Collection | Action | | Reasoning | A11y Info. | Fail Case |
|---|---|---|---|---|---|---|---|---|---|---|---|
| | | Grounding | Parsing | Action Pred. | | | GUI | API | | | |
| ScreenSpot | Human-designed | ✓ | ✗ | ✗ | 1,200+ | Human | N/A | N/A | ✗ | ✗ | ✗ |
| ScreenSpot-Pro | Human-designed | ✓ | ✗ | ✗ | 1,581 | Human | N/A | N/A | N/A | N/A | N/A |
| DeskVision | Online | ✓ | ✗ | ✗ | 54,855 | Auto. | N/A | N/A | ✗ | ✗ | ✗ |
| UI-Vision | Human-designed | ✓ | ✗ | ✓ | 8,227 | Human | ✓ | ✗ | ✗ | ✗ | ✗ |
| **GUI-360°** | **In-App/Online/ Search** | ✓ | ✓ | ✓ | **1,225,177** | Auto. | ✓ | ✓ | ✓ | ✓ | ✓ |

Prior datasets typically focus on a single aspect, rather than jointly enabling *GUI grounding*, *screen parsing*, and *action prediction* with execution traces and failure cases.

To fill these gaps, we introduce GUI-360°, a comprehensive dataset and benchmark suite for CUAs. GUI-360° is built around realism, scalability, and task breadth. Concretely, it provides the following:

1. **Real-world queries.** Task intents are collected from authentic sources (logs, forums, help content) to capture common user needs and frequent workflows.

2. **Automated pipeline.** An LLM-augmented pipeline handles template construction, instantiation, batched execution, and filtering, minimizing human effort while ensuring execution realism.

3. **Multi-task annotations.** Each example includes screenshots, accessibility metadata, intermediate agent "thoughts," and full trajectories (successes and failures). These support: (a) *GUI Grounding*, map a plan step to screen/UI; (b) *Screen Parsing*, enumerate interactable elements; (c) *Action Prediction*, given state and intent, predict the next action. See Fig. 1.

4. **Scale and coverage.** GUI-360° offers over 1.2M executed steps across thousands of trajectories on major Windows apps (Word, Excel, PowerPoint), with pipelines extendable to other software.

5. **Hybrid GUI+API space.** Reflecting modern CUA designs Zhang et al. (2025); Zhang et al., the action set combines GUI operations with API calls, enabling evaluation of both strategies.

Table 1 compares GUI-360° with existing GUI datasets across key dimensions. Unlike prior efforts that focus narrowly on grounding or small-scale scripted tasks, GUI-360° provides full coverage of grounding, parsing, and action prediction with a large-scale, automatically collected corpus. Notably, it is the first dataset to include accessibility information, reasoning supervision, and both GUI- and API-level actions, making it a uniquely comprehensive benchmark for CUA research.

To assess GUI-360°'s utility, we benchmark state-of-the-art vision–language models. Our results reveal consistent patterns: off-the-shelf models struggle with grounding in heterogeneous layouts and often fail in stepwise action prediction, leading to cascading errors. Training on GUI-360°-

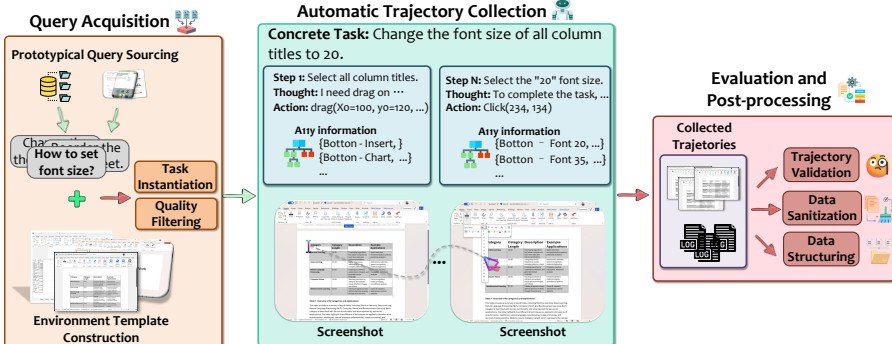

Figure 2: The data collection pipeline for GUI-360°.

Train yields significant gains. These findings underscore both the limitations of current models and the value of GUI-360° as a scalable, challenging benchmark for driving progress in CUAs.

## 2 RELATED WORK

**GUI and Computer-Using Agents**  LLMs have enabled agents Zhang et al. (2024a) that automate tasks across web Zheng et al. (2024; 2025a), mobile Wang et al. (2024b;a; 2025), and desktop platforms Zhang et al. (2024b; 2025); Qin et al. (2025). Desktop environments are particularly challenging for the CUAs due to high-resolution displays and complex layouts. Such agent typically rely on accessibility metadata or visual screen understanding Gur et al. (2023); Xie et al. (2023). When accessibility information is missing, they resort to screen parsing and visual grounding, as in SeeClick Cheng et al. (2024), OmniParser Lu et al. (2024), and GUI-Actor Wu et al. (2025). UFO Zhang et al. (2024b; 2025) pioneered hybrid approaches that combine accessibility with visual screen parsing, while recent efforts further integrate APIs for efficiency Zhang et al..

Ultimately, progress depends on large-scale data to support robust screen understanding and tool use. GUI-360° aims to fill this gap by providing a comprehensive dataset and benchmark.

**Data and Benchmarks for CUA**  A key obstacle in building effective CUAS lies in obtaining high-quality training data and reliable benchmarks. The desktop setting remains underexplored despite its greater complexity compared to web Deng et al. (2023); Zhou et al. (2023) and mobile domains Rawles et al. (2023; 2024). Several efforts have emerged in this space. UI-Vision Nayak et al. (2025) provides a human-annotated desktop benchmark with bounding boxes, UI labels, and action trajectories. DeskVision Xu et al. (2025) introduces a cross-OS dataset for desktop region captioning. However, these datasets and benchmarks either require costly human annotation, or cover only narrow subsets of tasks, limiting their scalability and diversity.

Our GUI-360°, addresses this gap by introducing an automated data collection pipeline that generates large-scale resources for GUI grounding, screen parsing, and action prediction. This design makes GUI-360° the most comprehensive and scalable dataset–benchmark suite to date for CUAs.

## 3 THE COLLECTION OF GUI-360°

The construction of GUI-360° follows a three-stage pipeline designed to maximize scalability while minimizing human effort, as shown in Figure 2. We begin by collecting *real-world* user queries from reliable sources to ensure coverage of diverse and realistic task intents. Then, we design a specialized CUA named TrajAgent to execute tasks in an automatic, consistent and high-quality manner. The TrajAgent generates and collect detailed trajectories that jointly support GUI grounding, screen parsing, and action prediction, enabling multi- Finally, we apply automated evaluators and systematic post-processing to verify correctness, filter noise, and enhance overall data quality.

### 3.1 QUERY ACQUISITION

High-quality and actionable user queries are the foundation of a reliable dataset, as they determine both task realism and execution fidelity Xu et al. (2025). To capture queries that reflect real-world

user needs, we design a dedicated acquisition pipeline (Figure 1), namely *(i)* **Prototypical Query Sourcing**, *(ii)* **Environment Template Construction**, *(iii)* **Task Instantiation**, and *(iv)* **Quality Filtering**. This four-stage pipeline ensures that the resulting queries are both realistic and executable.

**Prototypical Query Sourcing.** Most existing CUA datasets rely heavily on human-crafted or LLM-generated queries, which often fail to reflect real-world usage patterns or capture high-frequency tasks. To better ground our dataset in authentic user behavior, we source prototypical task descriptions from three complementary channels, namely *(i)* **In-App Help Content (In-APP)**, *(ii)* **Online Websites (Online)**, and *(iii)* **Search Queries (Search)**.The detailed description and statistics of the three sources are provided in Section A. These sources are real and reliable, in contrast to purely human-curated or LLM-synthetic data.

**Environment Template Construction.** Each user query must be grounded in a suitable environment that provides the necessary context for agent execution. For example, a task such as "make the first line of text bold" in Word requires a document containing editable text; without such a setup, the query is not actionable. Naively creating a bespoke environment for every query would be prohibitively expensive and redundant, since many queries share common contextual requirements.

To address this, we introduce an environment template construction process that systematically amortizes environment setup across queries. Specifically, we leverage GPT-4.1 to analyze each query and extract its underlying requirements (*e.g.*, the presence of text, a table, or an image). Queries with similar requirements are clustered and abstracted into environment template descriptions. We then manually instantiate a curated set of 66 high-frequency templates from these descriptions, which can support approximately **95%** of prototypical queries.

**Task Instantiation.** Prototypical queries collected from real-world sources are often vague and underspecified (*e.g.*, *"How to make text bold?"*). To be executable by an agent, each query must be grounded in a specific environment and reformulated into an actionable instruction (*e.g.*, *"Make the phrase 'Hello World' bold in the Word document"*). We refer to this process as *task instantiation*. To systematically achieve this, we design an automated two-stage pipeline. First, For each query, we provide the query with textual descriptions and screenshots of all available templates to GPT-4.1, to identify the most suitable template. Next, once a template is selected, the LLM rephrases the vague query into a fully instantiated one, grounded in the chosen environment. This pipeline is fully automated and enforces that all instantiated tasks are *actionable*, *concrete* and *scalable*.

**Quality Filtering.** Although task instantiation produces a large set of grounded queries, not all of them are suitable for reliable agent training. Many instantiated tasks may suffer from contextual mismatches, external dependencies, or inherent ambiguities. To ensure robustness, we design a *task filtering pipeline* that employs an GPT-4.1 as an automatic quality gate. The LLM-based judge evaluates each candidate task against a set of well-defined constraints, as detailed in Section C.

## 3.2 Automatic Trajectory Collection

Once queries have been instantiated and grounded, the next step is to generate corresponding action plans, execution trajectories, and outcomes. We develop a specialized execution framework, TrajAgent, that automatically completes queries in batch (Fig. 2) and records detailed execution data trajectories. The workflow consists of three main phases. First, For each task, TrajAgent initializes the corresponding environment template and ensures all preconditions. Then the agent performs the query step-by-step, generating a complete action trajectory while recording the full GUI state at each step. After execution, TrajAgent closes or resets the environment to a clean state, preparing it for the next task in the batch. This fully automated process ensures that every task is executed reliably, trajectories are captured consistently, and human intervention is entirely eliminated.

### 3.2.1 TrajAgent Design

The core requirement for large-scale trajectory collection is twofold: the executor must *(i)* complete instantiated queries with a high success rate to maximize data efficiency, and *(ii)* record every element required for dataset construction with strict fidelity. To meet these requirements we design

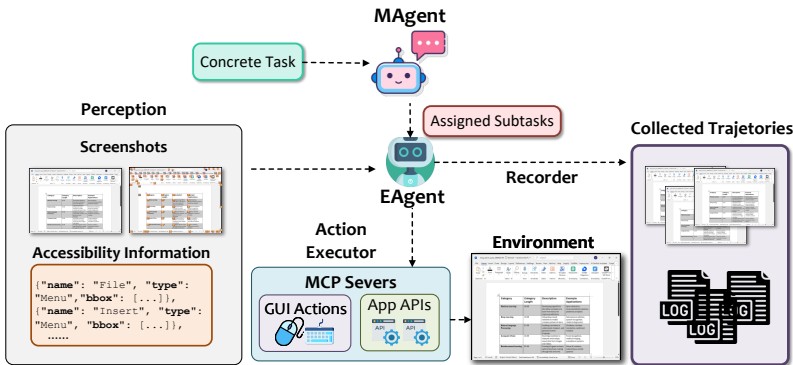

Figure 3: The overall architecture of the TrajAgent.

Table 2: Success rate of the two-stage execution strategy across applications.

|  | Word | Excel | PowerPoint | Total |
|---|---|---|---|---|
| Round 1 (GPT-4o) | 16.65% | 9.27% | 8.00% | 11.63% |
| Round 2 (GPT-4.1) | 16.03% | 18.20% | 14.91% | 16.38% |
| Overall | 30.50% | 25.78% | 21.71% | 26.09% |

TrajAgent, an orchestrated, multi-agent execution framework that reliably completes concrete tasks and produces high-fidelity execution traces suitable for downstream training and evaluation.

**Architecture overview.** TrajAgent follows a multi-agent orchestration pattern Zhang et al. (2024b) composed of a `MasterAgent` (MAgent), a pool of `ExecutionAgents` (EAgent), with a set of auxiliary services (Perception, Action Executor, and Recorder), as shown in Figure 3. The MAgent receives an concrete task and decomposes it into a sequence of manageable subtasks (planning). Each subtask is dispatched to an available EAgent for execution. EAgents operate as lightweight workers that (a) perceive the current GUI state, (b) select or synthesize the next low-level action (click/type/select/API call), (c) execute the action via UI automation or API invocation. The Recorder persistently logs the full GUI state and metadata before and after each action.

**Perception.** Within each EAgent, the Perception service captures a full-resolution screenshot at every decision step and queries the Windows accessibility API (UI Automation) Haverty (2005) to extract a list of actionable controls (name, type and exact bounding box). The accessibility-derived control metadata is rendered on the screenshot as a Set-of-Mark (SoM) Yang et al. (2023).

**Action Executor.** The Action Executor utilizes app-specific MCP servers Hou et al. (2025) to provide an extensible set of tools for the EAgent. In addition to GUI actions (*e.g.*, mouse clicks, keyboard input), our design incorporates app-level API actions, reflecting modern CUA practices Zhang et al.. These APIs improve task efficiency and serve as reliable fallbacks for error-prone GUI interactions. At each step, the EAgent selects the most appropriate action based on its observation, reasoning, and plan Yao et al. (2023). This iterative loop continues until the task is fully completed.

**Recorder.** The Recorder collects all multi-modal information at each execution step to construct the dataset. This includes screenshots, accessibility metadata, and agent outputs (thought, plan, *etc*). Importantly, a single execution of the agent produces data for all downstream tasks, significantly improving data efficiency. Table 11 in Section E summarizes the input and output for each task type.

**Two-Stage Execution.** To reduce dependency on the capabilities of a single model and improve coverage, we adopt a two-stage execution strategy. In the first stage, GPT-4o serves as the base model to complete the queries. Any queries that **fail in this stage** are then re-executed using a stronger model GPT-4.1. As shown in Table 2, the two-stage execution strategy substantially improves success rates compared to relying on a single model. Together, this staged approach boosts overall completion to 26.09%, showing that cascading models of complementary strengths increases both success rate and dataset versatility, while avoiding over-reliance on a single model.

Table 3: Statistics of GUI-360°-Train and GUI-360°-Bench (Successful-only).

| Metric | GUI-360°-Train | GUI-360°-Bench |
|---|---|---|
| Total Trajectories | 13,750 | 3,439 |
| Total Steps | 105,368 | 26,284 |
| Steps for Grounding Tasks | 79,487 | 19,780 |
| Steps for Screen Parsing | 105,368 | 26,284 |
| Steps for Action Prediction | 105,368 | 26,284 |
| Total Elements | 17,668,694 | 4,324,617 |
| Total Images | 210,736 | 52,568 |
| Average Elements per Image | 167.69 | 164.53 |
| GUI Action Rate (%) | 81.0 | 81.0 |
| API Action Rate (%) | 19.0 | 19.0 |

## 3.3 VALIDATION AND POST-PROCESSING

To ensure the high quality and usability of the automatically collected trajectories, we perform a three-stage post-processing procedure: *(i)* **Trajectory Validation.** Each trajectory is *automatically* evaluated by GPT-4.1 to retain only successful and executable task executions, ensuring that downstream training and evaluation are based on realistic completions. *(ii)* **Data Sanitization.** Low-quality steps, incomplete records, or any data that fail to meet predefined quality criteria are removed. This step eliminates noise and increases the overall reliability of the dataset. *(iii)* **Data Structuring.** The cleaned trajectories are reformatted and normalized into the required structure for the 3 downstream tasks. This includes standardizing screenshot metadata, accessibility information, action calls, and annotations to create a consistent, machine-readable dataset. For the *Action Prediction* task, we provide two input modalities: *visual-only* and *visual+a11y* setting. We present the detailed evaluation and post-processing in Section D, the data schema of GUI-360° in Section G.

## 3.4 GUI-360° STATISTICS

Following the pipeline described above, we construct a comprehensive dataset, GUI-360°-Train for training across three core GUI tasks, and a companion benchmark, GUI-360°-Bench, for evaluation.

**Scale.** Table 3 summarizes the statistics of GUI-360°-Train (80%) and GUI-360°-Bench (20%). In total, GUI-360° contains **13,750 trajectories** with over **105k steps**, averaging **7.66 steps per trajectory**. The dataset also provides **210k screenshots** paired with **17.7M annotated UI elements**, yielding rich multi-modal supervision at both the visual and accessibility levels. For evaluation, GUI-360°-Bench adds a further **3,439 trajectories** and **26k steps**, maintaining a similar distribution of average step length and GUI/API action rates. In addition, we include **62,170** trajectories comprising **1,093,525** steps for failure cases, which can serve as valuable signals for reinforcement learning Wang et al. (2024c). These failure cases capture challenging or error-prone situations that models often struggle with. Detailed statistics and analysis is shown in Section F.

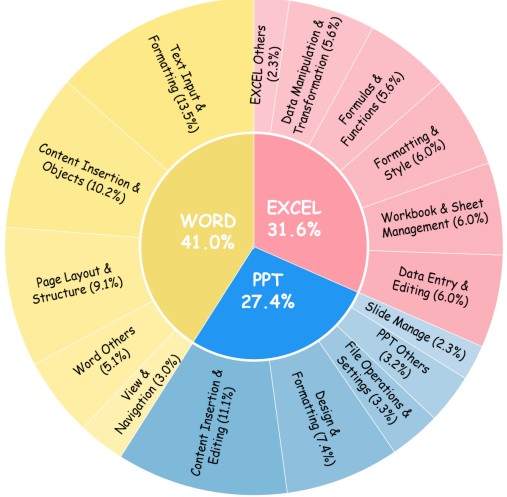

Figure 4: The **GUI-360° Composition**. For each app, tasks are divided into six categories.

This unprecedented scale, both in interaction traces and annotated elements, makes GUI-360° one of the largest and most comprehensive resources for GUI learning.

**Diversity.** Beyond scale, GUI-360° emphasizes breadth and functional diversity. Using GPT-4o, we classified user queries into fine-grained categories, as shown in Figure 4. The corpus spans Word

Table 4: Performance of different models on the GUI grounding task across applications.

| Model | Word | Excel | PowerPoint | Overall |
|---|---|---|---|---|
| GPT-4o | 15.22% | 5.08% | 5.41% | 9.38% |
| GPT-4.1 | 17.30% | 7.48% | 7.01% | 11.41% |
| o3 | 36.62% | 19.44% | 31.06% | 29.96% |
| GPT-5 | 34.52% | 20.31% | 17.36% | 25.34% |
| Qwen-2.5-VL-7B | 38.09% | 26.76% | 41.55% | 35.78% |
| UGround-7B | 57.44% | 43.09% | 59.53% | 53.85% |
| Aguvis-7B | 53.14% | 37.69% | 59.57% | 50.50% |
| UI-TARS-1.5 7B | 63.50% | 58.61% | 64.21% | 62.27% |
| GUI-Actor 7B | 54.84% | 45.84% | 62.68% | 54.50% |
| Qwen-2.5-VL-7B-SFT | 84.11% | 79.20% | **82.84%** | 82.30% |
| UI-TARS-1.5 7B-SFT | **84.73%** | **79.84%** | 81.98% | **82.49%** |

(41.0%), Excel (31.6%), and PowerPoint (27.4%), each with rich internal coverage. Word tasks range from text formatting to layout and review, Excel covers data entry, formulas, and visualization, while PowerPoint emphasizes content editing, design, and transitions. This balance ensures exposure to both frequent operations and rarer, long-tail behaviors, making it a comprehensive and challenging dataset and benchmark for CUA research.

## 4 EXPERIMENT

Our experimental evaluation proceeds in two stages. First, we perform an out-of-the-box evaluation of several state-of-the-art vision–language and agent models on GUI-360°-Bench. Second, we investigate how SFT on GUI-360°-Train improves model performance.

### 4.1 GUI GROUNDING

**Baselines & Metric.** We evaluate a mix of general-purpose and domain-specialized models. The general-purpose baselines are GPT-4o Hurst et al. (2024), GPT-4.1 OpenAI (2025b), o3 OpenAI (2025d), and GPT-5 OpenAI (2025c). We also include several open-source and grounding models: Qwen-VL-2.5 (7B) Bai et al. (2025), UGround-7B Gou et al. (2024), Aguvis-7B Xu et al. (2024), UI-TARS-1.5 (7B) Qin et al. (2025), and GUI-Actor (7B) Wu et al. (2025). Finally, we report results for SFT variants (Qwen-2.5 7B-SFT, UI-TARS-1.5 7B-SFT) that are trained on the GUI-360°-Train.

Predicted coordinates are evaluated against the accessibility-derived bounding box of the target element. More details about baselines and the metric are presented in Section H.1 and I.1.

**Performance Comparison.** Table 4 reports the evaluation results of different models on GUI-360°-bench for the GUI grounding task. We observe that general-purpose GPT models (*e.g.*, GPT-4o and GPT-4.1) achieve only modest performance, with overall accuracy below 12%. More advanced general models such as GPT-o3 and GPT-5 show improvements (20–30%), yet still struggle with precise GUI grounding. Domain-specific pretraining brings substantial gains: models like UGround-7B and GUI-Actor 7B surpass 50%, demonstrating the effectiveness of grounding-oriented pretraining. Finally, supervised fine-tuning (SFT) on GUI-360° yields the largest performance leap, with Qwen-2.5 7B-SFT and UI-TARS-1.5 7B-SFT achieving over 82% accuracy across applications. This progression clearly highlights the value of GUI-360° for both training and benchmarking: it not only reveals the limitations of general-purpose models on GUI tasks but also provides high-quality training data that enables fine-tuned models to achieve state-of-the-art performance.

### 4.2 SCREEN PARSING

**Baselines & Metric.** We evaluate both general VLMs and specialized screen-parsing models. General models include GPT-4o, GPT-4.1, GPT-o3, GPT-5, and Qwen-VL-2.5 (7B). Specialized baselines include OmniParser and OmniParser-v2 Lu et al. (2024).

Table 5: Overall comparison of different models on screen parsing.

| Model | Precision | Recall | F1 | Text Sim. | Avg IOU |
|---|---|---|---|---|---|
| GPT-4o | 0.034 | 0.014 | 0.019 | 0.147 | 0.229 |
| GPT-4.1 | 0.098 | 0.057 | 0.067 | 0.306 | 0.505 |
| o3 | 0.160 | 0.114 | 0.128 | 0.456 | 0.578 |
| GPT-5 | 0.111 | 0.080 | 0.089 | 0.304 | 0.569 |
| Qwen2.5-VL-7B | 0.181 | 0.010 | 0.015 | 0.113 | 0.211 |
| OmniParser | 0.411 | 0.459 | 0.406 | 0.565 | 0.731 |
| OmniParser v2 | **0.413** | **0.462** | **0.408** | **0.568** | **0.735** |

We evaluate screen parsing quality using three perspectives. First, we assess *detection accuracy* by reporting precision, recall, and F1. Second, we measure *localization quality* via mean intersection-over-union (IoU) between predicted and ground-truth bounding boxes. Finally, we evaluate *semantic correctness* by computing the average embedding similarity between predicted and ground-truth element names, assessing whether the agent recovers the intended role of each element. Definitions of these metrics are provided in the Section I.2.

**Performance Comparison.** Table 5 presents a detailed comparison of general-purpose VLMs and specialized screen parsing models across Word, Excel, and PowerPoint. Overall, general-purpose models such as GPT-4o, GPT-4.1, GPT-o3, and GPT-5 struggle with both element detection and localization, achieving low F1 scores and moderate mean IoU values. Notably, GPT-o3 exhibits the highest overall F1 among the general models and maintains relatively strong localization (IoU 0.578), but its recall remains limited, indicating many missed elements. GPT-4.1 and GPT-5 show uneven performance across applications: GPT-5 performs best on Excel but poorly on Power-Point, suggesting sensitivity to layout complexity and domain-specific content. In contrast, specialized parsers significantly outperform general-purpose models in all metrics. OmniParser and OmniParser-v2 achieve overall F1 scores above 0.40 and mean IoU above 0.73, with strong text similarity, demonstrating robust detection, accurate localization, and reliable semantic recovery.

These results reveal two key insights: *(i)* general-purpose VLMs are limited in screen parsing due to the need for fine-grained spatial reasoning and UI semantics, and *(ii)* task-specific training, as in OmniParser, provides substantial gains in both element coverage and semantic correctness, highlighting the necessity of specialized architectures for accurate and reliable screen understanding. More detailed breakdown and analysis per app domain are presented in Section J.

## 4.3 ACTION PREDICTION

**Baselines & Metric.** We evaluate a set of general-purpose VLMs, including GPT-4o, GPT-4.1, GPT-o3, GPT-5, and Qwen-VL-2.5 (7B). In addition, we fine-tune Qwen-VL-2.5 with GUI-360°-Train to examine post-training improvements. Experiments are conducted under two settings: with and without a11y information as input.

For action prediction, each step consists of three components: the *function* being invoked, its *arguments*, and a *status* flag indicating whether to continue or finish the trajectory. Accordingly, we report function accuracy (whether the predicted function matches the ground truth), argument accuracy (correctness of spatial or symbolic arguments given the function), and status accuracy (correctness of the status flag). We also measure an aggregated step success rate, which requires all three components to be correct simultaneously. Details are presented in Section I.3.

**Performance Comparison.** Table 6 reports the results of action prediction under the *visual-only* and *visual+a11y* settings. When relying on screenshots alone, all models perform poorly, with accuracy below 20% in most cases. This highlights the intrinsic difficulty of inferring precise action arguments purely from pixel-level cues, even for state-of-the-art proprietary VLMs such as GPT-4.1 and GPT-5. By contrast, providing accessibility information dramatically boosts performance. For example, GPT-4o improves from 3.12% to 36.71%, and GPT-4.1 nearly triples its performance from 2.82% to 39.19%. This demonstrates that structured element annotations effectively reduce the burden of visual grounding, enabling models to focus on action semantics.

Table 6: Model performance comparison with screen Visual-only (left) and with screen Visual+A11y (right). Values are reported as percentages.

| | **Visual-only** | | | | **Visual+A11y** | | | |
|---|---|---|---|---|---|---|---|---|
| Model | Word | Excel | PowerPoint | Total | Word | Excel | PowerPoint | Total |
| GPT-4o | 3.61 | 1.96 | 3.35 | 3.12 | **61.36** | 29.15 | 48.11 | 36.71 |
| GPT-4.1 | 3.60 | 1.88 | 2.55 | 2.82 | 35.46 | **33.13** | **50.98** | **39.19** |
| GPT-o3 | 16.85 | 13.06 | 24.42 | 17.92 | 34.00 | 28.76 | 48.84 | 36.72 |
| GPT-5 | 9.05 | 6.21 | 10.26 | 8.59 | 31.68 | 26.39 | 48.23 | 34.86 |
| Qwen-2.5 7B | 15.70 | 12.75 | 25.09 | 17.52 | 15.64 | 3.56 | 22.51 | 14.18 |
| Qwen-2.5 7B-SFT | **49.10** | **45.12** | **56.53** | **50.08** | 31.68 | 7.44 | 34.99 | 25.78 |

Table 7: Breakdown comparison of (Qwen-2.5-7B / SFT) models on action prediction.

| | w/o A11Y | | | | w/ A11Y | | | |
|---|---|---|---|---|---|---|---|---|
| Metric | Excel | Word | PowerPoint | Overall | Excel | Word | PowerPoint | Overall |
| Function Acc. | 61.7 / **81.1** | 63.0 / **81.2** | 88.5 / **91.0** | 69.8 / **83.9** | 50.6 / **80.6** | 66.6 / **83.2** | 90.8 / **91.5** | 69.0 / **84.8** |
| Args. Acc. | 12.8 / **45.4** | 15.8 / **49.4** | 25.2 / **56.7** | 17.6 / **50.3** | 3.6 / **7.5** | 15.7 / **31.8** | 22.6 / **35.2** | 14.3 / **25.9** |
| Status Acc. | **96.0** / 94.1 | **96.4** / 93.9 | **97.4** / 96.2 | **96.6** / 94.6 | 85.7 / **95.3** | **98.2** / 95.8 | **99.0** / 96.3 | 94.9 / **95.8** |
| Element Err. | 74.9 / **64.0** | 66.6 / **61.4** | 96.2 / **82.8** | 76.7 / **67.5** | 74.9 / **89.8** | 83.6 / **76.7** | 98.8 / **90.5** | 84.7 / **84.7** |

Furthermore, SFT on GUI-360° delivers substantial gains. Qwen-2.5 7B improves from 17.52% to 50.08% after SFT in the visual-only setting, a nearly threefold improvement, showing the strong training signal provided by GUI-360°. However, when a11y information is introduced, the benefits of SFT diminish, suggesting that a11y annotations already encode much of the structural alignment that SFT otherwise learns. Overall, these results highlight both the challenge and opportunity presented by GUI-360°: action prediction is extremely difficult without explicit structural information, but with accessibility-enhanced input and dataset-driven post-training, models can achieve substantial improvements.

**Breakdown Analysis.** To obtain a fine-grained understanding of action prediction, we break down the metric in Table 7. The *Element Err.* reflects ratio out-of-bounds coordinate predictions in the visual-only setting or incorrect element selections when a11y information is available.

We observe that SFT brings large gains in *function Acc.* and especially in *argument Acc.*, reducing errors by more than half across domains. The dominant source of failure remains *Args Mismatch Error*, which suggests that grounding actions to the correct interface element is the most challenging aspect. Notably, Element Err. contributes significantly to these mismatches, highlighting the model's difficulty in spatial grounding from raw screenshots.

Comparing the two evaluation settings, we find that introducing a11y metadata substantially reduces *Element Err.* errors, showing that structured semantic cues provide more reliable grounding than relying on visual information alone. Overall, these findings highlight that while models can correctly identify the intended operation, achieving precise grounding is still an open challenge, and incorporating structured UI metadata such as a11y is a promising direction.

## 5 CONCLUSION

We introduced GUI-360°, a large-scale dataset and benchmark for advancing CUAs. GUI-360° fills three critical gaps in the field: the lack of realistic task collections, the absence of scalable data collection pipelines, and the shortage of unified benchmarks spanning GUI grounding, screen parsing, and action prediction. Through an automated pipeline, we curated over 1.2M steps across thousands of trajectories in widely used Windows applications, paired with rich multimodal annotations including screenshots, accessibility metadata, reasoning traces, and both successful and failed executions. Our empirical evaluation highlights both the difficulty of the domain and the promise of GUI-360°. State-of-the-art vision–language models exhibit significant limitations when applied out-of-the-box, particularly in grounding and action prediction. Yet, fine-tuning on GUI-360° deliver consistent improvements, underscoring the dataset's utility as a training and evaluation resource.

ETHICS STATEMENT

We carefully considered ethical implications throughout the data collection, processing, and release pipeline. First, no human subjects were directly involved in the data collection process, and thus no personally identifiable information (PII) or sensitive user data is included. All queries and trajectories were generated and executed within controlled sandbox environments to ensure both privacy and security. Second, we adhered to software license terms and platform usage guidelines, ensuring that the collection process does not violate proprietary restrictions or legal compliance requirements. Third, we performed multiple post-processing stages, including trajectory validation, data sanitization, and structuring, to filter out incomplete, low-quality, or potentially misleading samples, thereby reducing the risk of harmful insights or erroneous model behaviors.

We acknowledge the potential downstream misuse of GUI automation technologies, such as unauthorized system manipulation or exploitation of accessibility features. To mitigate this, we restrict dataset release to non-sensitive application contexts (Word, Excel, PowerPoint) and exclude scenarios that could pose privacy, security, or safety risks. The dataset is intended solely for academic research on improving robustness, generalization, and evaluation of GUI agents. All models and baselines are evaluated under responsible-use guidelines, and we encourage future researchers to follow the same principles.

REPRODUCIBILITY STATEMENT

We place strong emphasis on reproducibility. To this end, we will release all data collection code, execution framework, and templates used to instantiate user queries. The full dataset GUI-360°, along with the benchmark split GUI-360°-Bench, will also be publicly available. Detailed descriptions of the pipeline are provided in Section 3, with task definitions summarized in Table 11, filtering statistics in Table 10, and additional implementation details, hyperparameters, and evaluation protocols are included in in Appendix K. Together, these resources ensure that both our dataset creation and experimental results can be fully reproduced, verified, and extended by the research community.

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

## A  QUERY SOURCES

To ensure both breadth and realism, we draw task intents from three complementary sources:

1. **In-App Help Content (In-App).** Built-in tutorials and official documentation provide standardized, structured descriptions of common workflows and functionalities. These queries cover canonical operations and ensure completeness of core task categories.

Table 8: Raw query statistics across applications and sources.

| Source | Word | Excel | PowerPoint |
|--------|------|-------|------------|
| In-App | 274 | 159 | 316 |
| Online | 1,914 | 3,393 | 1,701 |
| Search | 25,715 | 25,000 | 19,681 |
| Total | 27,903 | 28,552 | 21,698 |

2. **Online Websites (Online).** Community forums, Q&A platforms, and technical blogs supply diverse, naturally phrased queries. These contributions reflect real-world problem-solving scenarios and capture long-tail, user-specific needs often missing from official materials.

3. **Search Engine Queries (Search).** High-frequency queries collected from search engines highlight urgent and widely encountered issues. This source emphasizes practical challenges that users repeatedly face in everyday workflows.

Table 8 reports the distribution of raw queries across applications and sources. We observe that search data dominate the corpus, reflecting their naturally high frequency and broad coverage, while online forums contribute significant diversity, and in-app content provides structured coverage of baseline tasks. By combining these complementary sources, we achieve a balanced dataset that is both *authentic*—rooted in real user behavior—and *comprehensive*—covering canonical and long-tail workflows alike. This balance is crucial for building benchmarks that generalize beyond synthetic or hand-crafted examples.

## B  ACTION SET

To support diverse interaction across desktop applications, we design a unified action set that combines general-purpose GUI operations with application-specific APIs using MCP servers. The action set is deliberately lightweight yet expressive, enabling agents to cover the full spectrum of common productivity tasks while remaining tractable for model training.

The GUI actions (`click`, `type`, `drag`, `wheel_mouse_input`) form the foundation of interaction, as they are applicable to any graphical user interface. These actions abstract low-level mouse and keyboard events into structured calls, supporting variants such as absolute and normalized coordinates, modifier keys, and multi-step operations.

On top of this universal layer, we extend the action set with fine-grained APIs for **Word**, **Excel**, and **PowerPoint**. These APIs expose high-level document, spreadsheet, and presentation semantics, such as inserting tables, modifying cell values, reordering columns, adjusting font properties, or setting slide backgrounds. By combining GUI-agnostic operations with domain-specific APIs, the action set achieves both generality and efficiency: agents can rely on GUI actions for arbitrary interfaces while exploiting APIs for structured tasks where precise semantics matter.

Table 9 summarizes the full action set. This unified design allows agents trained on GUI-360° to operate seamlessly across heterogeneous applications, balancing robustness with expressivity.

## C  QUERY FILTERING ANALYSIS

1. **Non-Executable (`NONEXEC`)**: Inputs that do not describe a concrete action are removed. This includes subjective statements, vague preferences, or general inquiries that cannot be directly executed. For instance, instructions containing words such as "custom," "you want," or undefined operations like "edit text" without specifying the target element fall under this category.

2. **Cross-Application Dependency (`CROSSAPP`)**: Tasks that require interaction with applications beyond {app} are excluded. This includes operations that involve opening or manipulating content in external software (*e.g.*, Excel, Edge, File Explorer, or system settings). Representative examples include merging files across applications, printing documents (which requires printer integration), or exporting data to a third-party tool.

Table 9: Supported actions across Word, Excel, and PowerPoint, grouped by shared GUI actions and application-specific APIs.

| Action | Description | Type |
|---|---|---|
| click | Click at a given position (absolute or normalized), supporting left/right/middle/x button, single/double click, and optional modifier key. | GUI |
| type | Type text or hotkeys at a position, with options for clearing text or focusing on control. Supports special keys like {VK_CONTROL}c. | GUI |
| drag | Drag from a start to an end position with configurable mouse button, duration, and optional key hold (*e.g.*, shift, control). | GUI |
| wheel_mouse_input | Scroll at a given position with positive (up) or negative (down) wheel distance. | GUI |
| insert_table | Insert a table with a specified number of rows and columns. | Word API |
| select_text | Select exact text in the document. | Word API |
| select_table | Select a table by its index number. | Word API |
| select_paragraph | Select paragraphs by start and end indices, with option to restrict to non-empty ones. | Word API |
| save_as | Save the document with specified directory, file name, and extension (default: PDF). | Word API |
| set_font | Change font family and/or size. | Word API |
| table2markdown | Extract the contents of a worksheet table into Markdown format. | Excel API |
| insert_excel_table | Insert a table (list of lists) into a sheet at a specified starting cell. | Excel API |
| select_table_range | Select a range of cells by coordinates in a sheet. | Excel API |
| set_cell_value | Set the value (or formula) of a specific cell. | Excel API |
| auto_fill | Autofill values in a specified cell range. | Excel API |
| reorder_columns | Reorder columns in a sheet according to a given list of column names. | Excel API |
| set_background_color | Change the slide background color using a hex RGB value, for selected or all slides. | PowerPoint API |
| save_as | Save the presentation with specified directory, file name, and extension (default: PowerPointX). Optionally save slides as images. | PowerPoint API |

3. **Version Management (VERCTRL)**: Tasks that involve checking, updating, downgrading, or modifying the version of {app} are discarded. Since version management depends heavily on system environment and external factors, these tasks are not considered executable within the scope of our benchmark.

4. **Template Dependency (TPLMISS)**: Tasks that rely on specific document or workspace templates that are absent from the provided context are excluded. For example, instructions that assume the existence of a predefined table, chart, or object not available in the given file are categorized as TPLMISS. Note that application-wide settings (*e.g.*, enabling dark mode) do not fall under this restriction, as they do not depend on document templates.

5. **Irrelevant or Invalid (INVALID)**: Remaining cases that do not fit into the above categories, or are otherwise infeasible due to irrelevance, ambiguity, or context mismatch, are marked as invalid. For example, a task unrelated to {app} or lacking sufficient contextual information for execution would be discarded under this category.

Table 10 summarizes the results of our LLM-based task filtering across Word, Excel, and PowerPoint. Out of 79,075 candidate tasks, 59,553 (75.3%) are retained as NORMAL, indicating they are self-contained, concretely specified, and feasible given the provided templates and context. The remaining 24.7% of tasks are excluded due to various limitations. Overall, the filtering process ensures that the NORMAL subset comprises tasks that are well-specified, executable, and contextually grounded. This careful curation is critical for constructing a robust benchmark that reliably evaluates the performance of LLMs on single-application task execution, while also providing insights into common sources of task ambiguity or infeasibility.

Table 10: Distribution of task categories after filtering, shown as percentages of total candidate tasks for each application.

| Category | Word | Excel | PowerPoint |
|---|---|---|---|
| CROSSAPP | 15.61% | 13.27% | 12.13% |
| TPLMISS | 2.93% | 4.18% | 7.51% |
| NONEXEC | 1.56% | 1.68% | 3.65% |
| VERCTRL | 0.17% | 0.13% | 0.27% |
| INVALID | 2.70% | 6.98% | 1.46% |
| NORMAL | 77.02% | 73.76% | 74.97% |

## D  DETAILS OF EVALUATION AND POST-PROCESSING

**Trajectory Validation.**  To ensure the reliability of collected data, each trajectory undergoes automatic validation. We design an evaluation agent, *EvaAgent*, which leverages GPT-4.1 in an LLM-as-a-judge paradigm Gu et al. (2024). EvaAgent inspects the trajectory step by step, including the screenshot, accessibility information, executed actions, intermediate thoughts, and the final application state. Following prior work Wang et al. (2024c), it employs a chain-of-thought style reasoning process to decompose the query into several fine-grained evaluation criteria. A trajectory is marked as successful only if all criteria are satisfied, thereby enforcing a stricter notion of task completion.

To assess its reliability, we conducted a small-scale study on 100 randomly sampled trajectories. EvaAgent's judgments achieved 86% agreement with human annotators, demonstrating that it provides sufficiently accurate and scalable validation. Compared to hard-coded scripts or brittle oracle rules, this approach offers greater flexibility across diverse applications and enables rapid filtering of high-quality, successful trajectories at scale.

**Data Sanitization.**  After validation, we perform a final cleaning step to ensure completeness and consistency of the collected trajectories. This involves removing any step that lacks an executed action, a screenshot, or essential metadata required for downstream tasks. Such sanitization further improves the overall data quality and ensures that only fully executable, well-documented steps are retained for model training and evaluation.

**Data Structuring.**  Finally, we transform the sanitized data into a standardized JSON format tailored for model consumption, following the input–output specifications summarized in Table 11. For the *Action Prediction* task, we provide two input modalities: *visual-only* and *visual+a11y*:

- **Visual-only:** The model receives raw screenshots as input. Interaction-related arguments (*e.g.*, click positions) are represented as the absolute coordinates of the center of the corresponding bounding box.

- **Visual+a11y:** The model additionally receives the list of actionable elements from the accessibility API, which are also annotated on the screenshot using the Set-of-Mark (SoM) representation. Interaction arguments are expressed as element ID and name, chosen from the provided element list. This reduces the need for explicit coordinate prediction and lowers the visual grounding overhead.

Together, these stages guarantee that the final dataset is both high-quality and fully compatible with diverse CUA training and benchmarking pipelines.

For evaluation, we partition the trajectories into training (80%) and benchmark (GUI-360-Bench, 20%) splits. All three tasks—GUI Grounding, Screen Parsing, and Action Prediction—share the same data partition to maintain consistency across evaluations.

Table 11: Task input-output specifications for dataset collection.

| Task | Input | Output |
|------|-------|--------|
| GUI Grounding | Application screenshot, Agent's thought at the current step | Operation coordinates of the target element, obtained via accessibility APIs |
| Screen Parsing | Application screenshot | List of all actionable controls on screen with name and bounding box, *e.g.*, `{"name": "Open Menu", "bbox": [12,34,56,78]}` |
| Action Prediction | User query, Application screenshot, Accessibility information (optional) | Action call, with optional metadata such as agent's thought and plan |

## E  TASK INPUT-OUTPUT SPECIFICATIONS

Table 11 specifies the input-output design for dataset collection, providing a structured view of the three core tasks in our pipeline. Each task plays a distinct role in bridging raw user queries and screenshots with executable actions.

- **GUI Grounding.** This task focuses on localizing the precise element to interact with on the screen. By combining application screenshots with the agent's reasoning trace (*i.e.*, "thought"), the system determines the target coordinates of the actionable element. These coordinates are not heuristically guessed but are validated via accessibility APIs to ensure accurate alignment with system-recognized controls. This specification ensures high-fidelity mappings between natural instructions and the GUI surface.

- **Screen Parsing.** Screen parsing builds a structured representation of the current interface. Given only a screenshot, the system outputs a comprehensive list of actionable controls, including their semantic names and bounding boxes. This is critical for downstream tasks, as it transforms unstructured pixels into a machine-readable set of candidate actions. For example, a detected control may be represented as `{"name": "Open Menu", "bbox": [12,34,56,78]}`, enabling both grounding and prediction tasks to reason over well-defined objects instead of raw visual input.

- **Action Prediction.** This task connects user intent to system operations. The inputs are multi-modal, including the user's natural language query, visual context from the screenshot, and optional accessibility metadata. The output is an action call that specifies not only *what* operation to perform but may also include structured metadata such as the agent's intermediate plan or reasoning. This makes the action prediction task inherently more challenging and semantically rich, as it requires aligning natural language, visual layout, and accessibility data into an executable form.

Together, these tasks capture the full spectrum of dataset requirements: from raw perception (*screen parsing*) to grounding actions in concrete GUI elements (*GUI grounding*), and finally to bridging user intent with executable operations (*action prediction*). This decomposition ensures modularity, interpretability, and extensibility of the dataset for training and evaluation.

## F  DETAILED GUI-360° STATISTICS

Table 12 presents a detailed breakdown of the GUI-360° training and benchmark datasets across Word, Excel, and PowerPoint domains. Several key observations can be made:

**Scale and Balance.** The training split contains 13,750 trajectories and over 105k steps, roughly four times larger than the benchmark split (3,439 trajectories and 26k steps). This provides a rich training signal while ensuring that the evaluation set remains sufficiently large and diverse for reliable benchmarking. The average number of steps per trajectory is consistent across splits (7.66

Table 12: Training and test dataset statistics across domains (Word, Excel, PowerPoint).

| GUI-360°-Train | | | | |
|---|---|---|---|---|
| | Word | Excel | PowerPoint | Total |
| Total Trajectories | 5,633 | 4,348 | 3,769 | 13,750 |
| Total Steps | 41,742 | 29,363 | 34,263 | 105,368 |
| Average Steps per Trajectory | 7.41 | 6.75 | 9.09 | 7.66 |
| Steps for Grounding Tasks | 30,695 | 22,319 | 26,473 | 79,487 |
| Steps for Screen Parsing | 41,742 | 29,363 | 34,263 | 105,368 |
| Steps for Action Prediction | 41,742 | 29,363 | 34,263 | 105,368 |
| Total Elements | 3,270,104 | 12,211,852 | 2,186,738 | 17,668,694 |
| Total Images | 83,484 | 58,726 | 68,526 | 210,736 |
| Average Elements per Image | 78.34 | 415.89 | 63.82 | 167.69 |
| GUI Action Rate (%) | 76.1 | 80.7 | 87.4 | 81.0 |
| API Action Rate (%) | 23.9 | 19.3 | 12.6 | 19.0 |

| GUI-360°-Bench | | | | |
|---|---|---|---|---|
| | Word | Excel | PowerPoint | Total |
| Total Trajectories | 1,409 | 1,087 | 943 | 3,439 |
| Total Steps | 10,597 | 7,175 | 8,512 | 26,284 |
| Average Steps per Trajectory | 7.52 | 6.60 | 9.03 | 7.64 |
| Steps for Grounding Tasks | 7,784 | 5,444 | 6,552 | 19,780 |
| Steps for Screen Parsing | 10,597 | 7,175 | 8,512 | 26,284 |
| Steps for Action Prediction | 10,597 | 7,175 | 8,512 | 26,284 |
| Total Elements | 839,273 | 2,940,016 | 545,328 | 4,324,617 |
| Total Images | 21,194 | 14,350 | 17,024 | 52,568 |
| Average Elements per Image | 79.20 | 409.76 | 64.07 | 164.53 |
| GUI Action Rate (%) | 76.2 | 80.2 | 87.5 | 81.0 |
| API Action Rate (%) | 23.8 | 19.8 | 12.5 | 19.0 |

in training vs. 7.64 in benchmark), indicating that both datasets preserve similar task complexity distributions.

**Task Coverage.** Each trajectory decomposes into multiple step-level tasks, spanning GUI grounding, screen parsing, and action prediction. For both splits, every step contributes to screen parsing and action prediction, while a large subset (79.5k in training and 19.8k in benchmark) is specifically dedicated to grounding tasks. This balanced design ensures that models trained on GUI-360° are exposed to diverse subproblems critical for end-to-end agent performance.

**Domain Differences.** Excel trajectories contain significantly more UI elements per image (415.9 in training and 409.8 in benchmark) compared to Word (78.3/79.2) and PowerPoint (63.8/64.1). This highlights Excel's inherent interface complexity and motivates the need for robust grounding and parsing strategies in grid-heavy environments. By contrast, PowerPoint tasks involve fewer elements but slightly longer trajectories (around 9 steps on average), reflecting the multi-step nature of slide composition tasks.

**Action Modality.** The ratio of GUI-based to API-based actions remains stable across splits and applications. On average, 81% of actions are executed via GUI operations, while 19% rely on APIs. Interestingly, PowerPoint shows a higher reliance on GUI actions (87%), while Word and Excel involve more mixed usage. This suggests that real-world GUI agents must flexibly support both interaction modalities to achieve broad coverage.

**Implications.** Overall, GUI-360° offers large-scale, well-balanced supervision for GUI agent training. Its fine-grained decomposition into tasks, diversity across domains, and preservation of modality ratios ensure that models can be both trained and evaluated under realistic and representative conditions.

## G   GUI-360° Schema

Each execution step in GUI-360° is stored as a structured JSON object following a unified schema. This schema ensures consistency across tasks and provides rich multimodal supervision for grounding, parsing, and action prediction. Table 13 summarizes the key fields.

Table 13: Execution Step Schema for GUI-360°. Each entry records metadata, screenshots, accessibility data, reasoning traces, and actions.

| Field | Description |
|---|---|
| execution_id | Unique identifier for the execution instance (*e.g.*, word_1_1). |
| app_domain | Application domain (*e.g.*, Word, Excel, PowerPoint). |
| request | Natural-language task description provided to the agent. |
| template | Environment template file used for instantiation. |
| step_id/total_steps | Current step index and total number of steps in the trajectory. |
| evaluation | Automatic assessment of the step, including reasoning, evidence, sub-scores, and a final completeness label. |
| step.screenshots | Multiple synchronized screenshots: clean view, full desktop, annotated version, and selected-controls view. |
| step.ui_tree | Hierarchical UI structure with element IDs, names, control types, bounding boxes, and children. |
| step.control_infos | Metadata from accessibility APIs and merged control sources, providing bounding boxes, labels, and semantic text. |
| step.observation | Agent's textual observation of the current state. |
| step.thought | Agent's intermediate reasoning for the next action. |
| step.action | Executed action, including function type (*e.g.*, click), arguments, target coordinates, and status flag. |
| status | Overall status of the step (CONTINUE or FINISH). |

**Discussion.**   The schema integrates three complementary perspectives: (i) **Visual context** through multi-view screenshots, (ii) **Structural context** via accessibility metadata and hierarchical UI trees, and (iii) **Cognitive traces** through observations, reasoning, actions, and evaluations. This rich structure allows GUI-360° to jointly support grounding, parsing, and action prediction, while enabling both supervised training and fine-grained evaluation. By standardizing every execution step, the schema provides a scalable foundation for reproducibility and extensibility across applications.

## H   Baseline Details

We summarize the baselines evaluated on GUI-360° across the three core tasks: GUI grounding, screen parsing, and action prediction. These baselines include both general-purpose vision–language models and domain-specific approaches designed for GUI reasoning. Below we briefly introduce each group of models.

### H.1   GUI Grounding

• **GPT-4o** Hurst et al. (2024): proprietary multimodal VLM used off-the-shelf for grounding.

• **GPT-4.1** OpenAI (2025b): proprietary VLM emphasizing instruction-following and tool use.

• **o3** OpenAI (2025d): OpenAI "reasoning" model family; evaluated zero-shot for grounding.

- **GPT-5** OpenAI (2025c): latest OpenAI flagship; strong general reasoning baseline.

- **Qwen2.5-VL-7B** Bai et al. (2025): open-source 7B multimodal baseline.

- **UGround-7B** Gou et al. (2024): GUI visual grounding model (Qwen2-VL backbone) trained with large-scale GUI data.

- **Aguvis-7B** Xu et al. (2024): vision-centric GUI agent with a unified cross-platform action space.

- **UI-TARS-1.5 (7B)** Qin et al. (2025): multimodal agent optimized for GUI reasoning and interactive tasks.

- **GUI-Actor (7B)** Wu et al. (2025): coordinate-free grounding model with an attention-based action head.

- **SFT variants**: Qwen2.5-VL-7B fine-tuned on GUI-360° for task-adapted grounding.

## H.2 SCREEN PARSING

- **GPT-4o / GPT-4.1 / o3 / GPT-5** Hurst et al. (2024); OpenAI (2025b;d;c): general-purpose VLMs used off-the-shelf for detection/localization.

- **Qwen2.5-VL-7B** Bai et al. (2025): open-source multimodal baseline.

- **OmniParser / OmniParser-v2** Lu et al. (2024): screen-parsing tools that produce element sets (names + bounding boxes) from raw screenshots.

## H.3 ACTION PREDICTION

- **GPT-4o / GPT-4.1 / o3 / GPT-5** Hurst et al. (2024); OpenAI (2025b;d;c): proprietary VLMs evaluated in both *visual-only* and *visual+a11y* settings.

- **Qwen2.5-VL-7B** Bai et al. (2025): open-source baseline for structured action generation.

- **Qwen2.5-VL-7B-SFT**: supervised fine-tuning on GUI-360° for step-wise function/argument/status prediction.

**Summary.** Together, these baselines span a spectrum from general-purpose VLMs to specialized GUI-focused agents. This diversity allows us to systematically evaluate the unique challenges posed by GUI-360° across grounding, parsing, and action prediction, and to measure how far current models remain from robust, human-level computer-using agents.

# I PERFORMANCE METRICS

## I.1 GUI GROUNDING

The primary evaluation metric for GUI grounding is **accuracy**, defined as the proportion of predictions where the predicted coordinate $\hat{c}_i$ lies within the bounding box of the corresponding ground-truth target element $b_i$. Formally,

$$\text{Acc} = \frac{1}{N} \sum_{i=1}^{N} \Vdash\{\hat{c}_i \in b_i\},$$

where $N$ is the total number of test cases and $\Vdash\{\cdot\}$ is the indicator function.

## I.2 SCREEN PARSING

We measure parsing quality along three complementary axes: (i) element detection accuracy (precision / recall / F1), (ii) localization quality (mean IoU on matched pairs), and (iii) semantic name accuracy (average text embedding similarity on matched pairs). All metrics are computed per image and then averaged across the benchmark (macro-average).

Let $G$ be the ground-truth set of elements for an image and $P$ the predicted set. We obtain a one-to-one matched set $M \subseteq P \times G$ by performing greedy bipartite matching sorted by descending IoU,

and keeping only pairs with $\text{IoU} > 0.5$. For a predicted box $p$ and ground-truth box $g$ we use the standard intersection-over-union:

$$\text{IoU}(p, g) \;=\; \frac{\text{area}(p \cap g)}{\text{area}(p \cup g)}.$$

For each image $i$ define:

$$\text{Precision}_i \;=\; \frac{|M_i|}{|P_i|}, \qquad \text{Recall}_i \;=\; \frac{|M_i|}{|G_i|}, \qquad \text{F1}_i \;=\; \frac{2 \cdot \text{Precision}_i \cdot \text{Recall}_i}{\text{Precision}_i + \text{Recall}_i},$$

where $|\cdot|$ denotes set cardinality. The reported precision, recall, and F1 are the macro-averages across images:

$$\text{Precision} \;=\; \frac{1}{N} \sum_{i=1}^{N} \text{Precision}_i, \quad \text{Recall} \;=\; \frac{1}{N} \sum_{i=1}^{N} \text{Recall}_i, \quad \text{F1} \;=\; \frac{1}{N} \sum_{i=1}^{N} \text{F1}_i.$$

To quantify localization quality, we compute the mean IoU for each image $i$ over the matched pairs $M_i$. If an image has no matched pairs ($|M_i| = 0$), we define its IoU as 0. The overall mean IoU is then the macro-average across all images:

$$\overline{\text{IoU}} \;=\; \frac{1}{N} \sum_{i=1}^{N} \begin{cases} \frac{1}{|M_i|} \sum_{(p,g) \in M_i} \text{IoU}(p, g), & |M_i| > 0 \\ 0, & |M_i| = 0 \end{cases}.$$

Similarly, for semantic-name accuracy, we embed predicted and ground-truth names using a sentence encoder $\phi(\cdot)$ (*i.e.*, a sentence-transformer Reimers & Gurevych (2019)) and compute the cosine similarity for each matched pair. If an image has no matches, we assign a similarity of 0 for that image. The macro-average over all images is:

$$\overline{\text{Sim}} \;=\; \frac{1}{N} \sum_{i=1}^{N} \begin{cases} \frac{1}{|M_i|} \sum_{(p,g) \in M_i} \frac{\langle \phi(\text{name}_p), \phi(\text{name}_g) \rangle}{\|\phi(\text{name}_p)\| \, \|\phi(\text{name}_g)\|}, & |M_i| > 0 \\ 0, & |M_i| = 0 \end{cases}.$$

Together, these metrics separate *whether* elements are detected (precision/recall/F1), *how precisely* they are localized (mean IoU), and *how well* their semantic roles are recovered (mean embedding similarity).

### I.3 ACTION PREDICTION

The evaluation of action prediction is more nuanced than grounding, since each action step is composed of a *function*, a set of *arguments*, and a *status* flag (continue or finish). We therefore report three component accuracies and one aggregated metric:

- **Function accuracy** ($\text{Acc}_{\text{func}}$): the proportion of predictions where the predicted function $\hat{f}_i$ exactly matches the ground-truth function $f_i$.

$$\text{Acc}_{\text{func}} = \frac{1}{N} \sum_{i=1}^{N} \mathbb{1}\{\hat{f}_i = f_i\}.$$

- **Argument accuracy** ($\text{Acc}_{\text{args}}$): evaluated conditionally on the predicted function. If the function is a spatial action such as `click`, correctness requires that the predicted coordinate $(\hat{x}_i, \hat{y}_i)$ falls inside the ground-truth bounding box $b_i$. For symbolic arguments (*e.g.*, menu item name, keystroke, value,), correctness requires exact match between predicted and ground-truth arguments. Formally,

$$\text{Acc}_{\text{args}} = \frac{1}{N} \sum_{i=1}^{N} \mathbb{1}\{\hat{a}_i \equiv a_i \mid f_i\},$$

where the equivalence relation $\equiv$ depends on the function type $f_i$.

Table 14: Comparison of different models across domains (Precision, Recall, F1, Text Similarity, Avg IOU Accuracy).

| Model | Domain | Precision | Recall | F1 | Text Sim. | Avg IOU |
|---|---|---|---|---|---|---|
| GPT-4o | Word | 0.040 | 0.017 | 0.024 | 0.170 | 0.252 |
| | Excel | 0.020 | 0.002 | 0.004 | 0.085 | 0.133 |
| | PowerPoint | 0.037 | 0.021 | 0.026 | 0.171 | 0.282 |
| | **Overall** | 0.034 | 0.014 | 0.019 | 0.147 | 0.229 |
| GPT-4.1 | Word | 0.101 | 0.065 | 0.077 | 0.307 | 0.518 |
| | Excel | 0.102 | 0.026 | 0.039 | 0.278 | 0.514 |
| | PowerPoint | 0.091 | 0.073 | 0.080 | 0.330 | 0.480 |
| | **Overall** | 0.098 | 0.057 | 0.067 | 0.306 | 0.505 |
| o3 | Word | 0.173 | 0.128 | 0.144 | 0.481 | 0.631 |
| | Excel | 0.178 | 0.099 | 0.118 | 0.335 | 0.523 |
| | PowerPoint | 0.129 | 0.109 | 0.115 | 0.526 | 0.559 |
| | **Overall** | 0.160 | 0.114 | 0.128 | 0.456 | 0.578 |
| GPT-5 | Word | 0.106 | 0.079 | 0.088 | 0.315 | 0.615 |
| | Excel | 0.172 | 0.109 | 0.126 | 0.274 | 0.574 |
| | PowerPoint | 0.065 | 0.056 | 0.059 | 0.314 | 0.508 |
| | **Overall** | 0.111 | 0.080 | 0.089 | 0.304 | 0.569 |
| Qwen2.5-VL-7B | Word | 0.384 | 0.014 | 0.023 | 0.137 | 0.358 |
| | Excel | 0.041 | 0.002 | 0.003 | 0.082 | 0.094 |
| | PowerPoint | 0.047 | 0.011 | 0.014 | 0.111 | 0.128 |
| | **Overall** | 0.181 | 0.010 | 0.015 | 0.113 | 0.211 |
| OmniParser | Word | 0.392 | 0.520 | 0.440 | 0.619 | 0.730 |
| | Excel | 0.431 | 0.217 | 0.270 | 0.450 | 0.748 |
| | PowerPoint | 0.417 | 0.588 | 0.479 | 0.594 | 0.718 |
| | **Overall** | 0.411 | 0.459 | 0.406 | 0.565 | 0.731 |
| OmniParser v2 | Word | 0.396 | 0.525 | 0.444 | 0.625 | 0.738 |
| | Excel | 0.431 | 0.217 | 0.270 | 0.450 | 0.748 |
| | PowerPoint | 0.418 | 0.590 | 0.481 | 0.596 | 0.721 |
| | **Overall** | 0.413 | 0.462 | 0.408 | 0.568 | 0.735 |

- **Status accuracy** ($\text{Acc}_{\text{status}}$): whether the predicted status flag $\hat{s}_i$ matches the ground-truth status $s_i$.

- **Step success rate** ($\text{Acc}_{\text{step}}$): a step is considered correct only if all three components (function, arguments, status) are correct simultaneously.

$$\text{Acc}_{\text{step}} = \frac{1}{N} \sum_{i=1}^{N} \mathbb{1}\{\hat{f}_i = f_i \wedge \hat{a}_i \equiv a_i \wedge \hat{s}_i = s_i\}.$$

## J  PER-APPLICATION ANALYSIS ON SCREEN PARSING

We further analyze model performance across applications (detailed results in Appendix 14). On **Word**, which features text-rich but visually ambiguous layouts, language models benefit from strong semantic understanding, yielding relatively high text similarity, yet they struggle with accurate localization, often predicting correct labels with shifted bounding boxes. In contrast, OmniParser achieves consistently higher recall and IoU, indicating the importance of GUI-aware geometric priors. **Excel** proves to be the most challenging domain due to its high-density and repetitive grid structure. Most general-purpose models collapse in recall, missing a large fraction of small, homogeneous controls. OmniParser maintains moderate recall and strong IoU by leveraging layout-aware parsing, highlighting that specialized table-structure modules are critical for handling spreadsheet interfaces. In **PowerPoint**, where controls are visually salient and layouts more regular, models perform relatively well. OmniParser attains both high recall and accurate localization, and even general

VLMs perform better than in Excel, though complex objects like charts and grouped components remain difficult. Overall, these findings suggest that GUI-specific inductive biases and structure-aware detection are essential, particularly for dense spreadsheet environments, while semantic recovery alone is insufficient without precise localization.

## K    IMPLEMENTATION DETAILS

**Data Collection.**    To construct GUI-360°, we deployed a cluster of **15 Windows 11 virtual machines**, each provisioned with 4 CPU cores. These VMs executed tasks in parallel, enabling efficient large-scale trajectory collection. Task execution followed a two-phase strategy: in **Phase 1**, GPT-4o was used as the agent; in **Phase 2**, all failed tasks were re-executed with GPT-4.1 for recovery (see Section 3). Both models were queried with the temperature fixed at `0.0` to ensure deterministic outputs and reproducibility.

**Model Access.**    For evaluation on GUI-360°-Bench, we used multiple OpenAI models, including GPT-4o, GPT-4.1, o3, and GPT-5, all accessed via the **Azure OpenAI Service**. Unless otherwise stated, the decoding temperature was set to `0.0` across all experiments to minimize variance and ensure consistent evaluation.

**Fine-tuning and Training.**    All supervised fine-tuning (SFT) experiments were conducted on a compute cluster equipped with **NVIDIA A100 GPUs** (40GB memory per GPU), finetuned with 1 epoch. Specifically, each run was distributed across 4 A100 GPUs using mixed-precision training (FP16) for efficiency. We adopted standard optimization settings following prior work on multimodal fine-tuning, with learning rates tuned over $\{1e\text{-}5, 5e\text{-}6, 1e\text{-}6\}$. Checkpointing and gradient accumulation were applied to ensure stable training for long trajectories.

## LLM USAGE STATEMENT

In preparing this work, we used large language models (LLMs) strictly as an assistive tool for text polishing and minor language refinement. All research ideas, technical designs, analyses, and conclusions were conceived and carried out entirely by the authors.

