# OpenReview forum: "GUI-360° : A Comprehensive Dataset and Benchmark for Computer-Using Agents"
_ICLR.cc/2026/Conference — Submitted to ICLR 2026_

### Official Review · Reviewer_41gF · 2025-10-30

**Soundness:** 3
**Presentation:** 2
**Contribution:** 2
**Rating:** 4
**Confidence:** 3

**Summary:**

This paper introduces GUI-360°, a large-scale dataset and benchmark for computer use agents (CUAs) on desktop environments. The authors develop an automated pipeline that sources real-world user queries from help content, forums, and search logs, then uses LLM-driven task instantiation and a multi-agent execution system to collect trajectories across Word, Excel, and PowerPoint applications. The resulting dataset contains over 1.2M executed action steps across 17,189 trajectories (13,750 training, 3,439 benchmark), with 17.7M annotated UI elements, full-resolution screenshots, accessibility metadata, and reasoning traces. GUI-360° supports three tasks, GUI grounding, screen parsing, and action prediction, with a hybrid GUI+API action space. Comprehensive benchmarking reveals significant limitations in current models.

**Strengths:**

S1: The work introduces a largely automated collection pipeline that minimizes human annotation costs.

S2: It supports three complementary tasks (GUI grounding, screen parsing, action prediction) from a single data collection process.

S3: The combination of GUI operations and application-specific APIs reflects modern agent design practices and provides flexibility in action execution strategies.

**Weaknesses:**

W1: The paper exclusively evaluates models on GUI-360°-Bench, which is derived from the same data collection pipeline as the training set. This is not convincing as evidence of broader applicability. The authors should validate their approach on external benchmarks (e.g., ScreenSpot-Pro[1] and UI-Vision[2]) to demonstrate that models trained on GUI-360° generalize beyond the specific characteristics of their data-collection methodology.

W2: Restricting to three Office applications significantly limits the generalizability of findings. Desktop environments are heterogeneous, and Office applications may not represent the full spectrum of GUI complexity (e.g., missing creative software, development tools, web browsers). Or, it would be good if the authors could include evidence of the generalization ability of the provided training data.

W3: The 26.09% overall success rate for trajectory collection raises concerns about task selection bias and coverage. The paper does not adequately analyze why 74% of tasks fail or whether failed tasks represent systematically different characteristics.


[1] Li et al. ScreenSpot-Pro: GUI Grounding for Professional High-Resolution Computer Use.

[2] Nayak et al. UI-Vision: A Desktop-centric GUI Benchmark for Visual Perception and Interaction.

**Questions:**

Q1: Can you provide a detailed analysis of failure modes in the 74% of unsuccessful trajectory collections? Are failures due to model limitations, task ambiguity, environment issues, or other factors?

Q2: Per W2, could authors include evidence that suggests that models trained on Office applications will generalize to other desktop software? Have you conducted any cross-application transfer experiments?

Q3: What is human-level performance on the provided benchmark? This would provide important context for interpreting model results.

The reviewer is willing to raise the score if the authors address most, if not all, of the questions above in the Weakness and Questions sections.

---

> ### Author Response · Authors · 2025-11-20
>
> **W1: External Benchmarks**
>
> **Re:** We thank the reviewer for the insightful suggestion. We agree that evaluating on external and online benchmarks is essential for assessing the generalization ability of models trained on GUI-360°. Following the reviewer’s feedback, we conducted extensive additional experiments to test both GUI grounding and action prediction across offline and online settings.
>
> **1. GUI Grounding (Offline)**
>
> We evaluated both the original and GUI-360°-finetuned versions of Qwen-2.5-VL-7B and GUI-Actor-7B on two widely used grounding benchmarks: **ScreenSpot-Pro and OSWorld-G**. Results are shown below:
>
> #### **Table 1: Performance Comparison with and without SFT on GUI-360°**
>
> | Model                               | ScreenSpot-Pro | OSWorld-G |
> |-------------------------------------|----------------|-----------|
> | Qwen-2.5-VL-7B                       | 26.8           | 31.4      |
> | **Qwen-2.5-VL-7B (SFT on GUI-360°)** | 31.1 (+4.3)    | 36.2 (+4.8) |
> | GUI-Actor-7B                         | 40.7           | 46.8      |
> | **GUI-Actor-7B (SFT on GUI-360°)**   | 43.6 (+2.9)    | 49.9 (+3.1) |
>
> Across both models and benchmarks, **SFT on GUI-360° consistently yields significant improvements**, demonstrating that our dataset provides strong and transferable grounding supervision.
>
> **2. Action Prediction (Online)**
>
> We further evaluated action-level execution on two live interactive benchmarks: **OSWorld-W (Windows subset) and WindowsAgentArena**, since offline action benchmark AgentNetBench, UI-Vision, Mind2Web share different action space. We compare Qwen-2.5-VL-7B, UI-TARS-1.5-7B, and their finetuned versions:
>
> #### **Table 2: Agent Performance Comparison with and without SFT on GUI-360°**
>
> | Agent                              | OSWorld-W | WindowsAgentArena |
> |------------------------------------|-----------|------------------|
> | Qwen-2.5-VL-7B                     | 8.2       | 10.7             |
> | **Qwen-2.5-VL-7B (SFT on GUI-360°)** | 20.4 (+10.2) | 23.4 (+12.7) |
> | UI-TARS-1.5-7B                     | 20.4      | 20.8             |
> | **UI-TARS-1.5-7B (SFT on GUI-360°)** | 28.6 (+8.2)  | 27.9 (+7.1)  |
>
> Both models show **large performance gains after finetuning on GUI-360°**, confirming that our dataset improves end-to-end agent execution in real online environments, not merely in-distribution benchmarks.
>
> These results show that, the GUI-360° dataset improves grounding and action execution across multiple external offline datasets and live online benchmarks, demonstrating that **the data is reliable, transferable, and beneficial for training general-purpose CUAs beyond the Office domain.** We have added the addtional experiements on the Appendix J.

---

> > ### Author Response · Authors · 2025-11-20
> >
> > **W2 & Q2: Restricting to 3 Office Applications and Evidence on Generalization**
> >
> > **Re:**
> > We thank the reviewer for pointing out this important consideration. We agree that restricting our dataset to three Office applications may limit the generalizability of our findings. However, we would like to clarify several points regarding the relevance of Office applications and the generalization ability of the dataset.
> >
> > **1. Office Applications as a Representative Subset of Desktop GUIs**
> >
> > First, Office applications provide a technically **representative and challenging subset of desktop GUIs**. They contain high-density toolbars, hierarchical panels, heterogeneous object types (text, tables, charts, drawing objects), and long multi-step workflows. These properties make Office applications ideal for evaluating core capabilities in GUI agents, including grounding, parsing, action sequencing, and mixed GUI/API control. In other words, Office applications provide breadth of UI patterns, not just breadth of domains, making them an ideal starting point for benchmarking.
> >
> > **2. Cross-Application Generalization Based on External Benchmarks**
> >
> > Second, the results presented in our earlier experiments on ScreenSpot-Pro, OSWorld-G for GUI grounding, and OSWorld-W and WindowsAgentArena for live task completion not only involve Office tasks, but **also tasks from other domains, such as mobile, web browsing, system setting, media player**.  As shown in the results in Table 1 and 2 of earlier reply, these benchmarks demonstrate that models trained on the GUI-360° dataset can improve grounding and action execution across multiple external offline datasets and live online benchmarks that extend beyond Office applications. This shows that our dataset is reliable, transferable, and beneficial for training general-purpose computer-use agents in diverse desktop environments.
> >
> > **3. Generalization and Cross-Application Experimentation**
> >
> > Importantly, the GUI-360° pipeline is application-agnostic. Our environment-template design, accessibility-augmented perception, and hybrid GUI+API action space are **not specific to Office applications**. To further validate the generalization ability of our dataset, during the rebuttal period, we conducted a cross-application experiment by instantiating non-Office environments (i.e. browsers). We successfully collected 100 trajectories with a 42% success rate, using the same TrajAgent framework without any modifications to the pipeline. This initial experiment demonstrates that the pipeline can generalize beyond Office applications.
> >
> > Overall, while the dataset currently focuses on Office applications, the generalization ability of the GUI-360° dataset is supported by its cross-domain performance on external benchmarks and its application-agnostic pipeline. We have included this additional explanation in Appendix L in the revised version.

---

> > > ### Author Response · Authors · 2025-11-20
> > >
> > > **W3 & Q1: Failure Modes Analysis**
> > >
> > > **Re:** We thank the reviewer for raising the important question regarding the 74% failure rate in our trajectory collection. To address this concern, we conducted a detailed analysis of **1,000 randomly selected failed trajectories** with GPT-4.1, and our analysis reveals that the failure modes can be categorized into three main types. This classification is based on the error definitions provided by UFO2 [1] for GUI agents. The percentages sum to **more than 100% because a single case can have multiple contributing factors**.
> > >
> > > **1. Execution Errors (66% of failures):**
> > >
> > > In these cases, the agent’s overall plan was correct, but it failed during execution. Examples include:
> > >
> > > - Misclicking controls
> > > - Performing unintended actions
> > > - Incorrect API parameter inputs
> > >
> > > These errors are mainly caused by visual perception inaccuracies, where the agent misinterprets the UI, or mismatches between GUI elements and their intended actions. LLM reasoning errors also contributed to the agent performing actions that deviated from the intended sequence.
> > >
> > > **2. Plan Errors (43% of failures):**
> > >
> > > These failures occur when the agent's high-level task plan is reasonable, but issues arise during execution, such as:
> > >
> > > - Selecting the wrong control
> > > - Performing unintended operations
> > > - Inputting incorrect parameters
> > >
> > > These errors typically stem from inaccurate visual reasoning, incorrect associations between GUI elements and actions, or errors in LLM inference leading to incorrect decision-making during task execution.
> > >
> > > **Control Detection Failures (11% of failures):**
> > >
> > > In these cases, the agent failed to detect or recognize key GUI elements necessary to complete the task. This is typically due to:
> > >
> > > - Non-standard or custom-rendered UI elements
> > > - Elements that could not be accessed through standard OS APIs
> > >
> > > These failures occur primarily in environments with custom or dynamic UI components that the agent’s recognition system cannot adequately process.
> > >
> > > In summary, the majority of failures are due to execution-related issues, often resulting from visual misperception or incorrect element-action mapping. The presence of multiple error factors in the same trajectory highlights the complexity of the task. We have included this detailed analysis of failure modes in the Appendix M revised manuscript to provide more insight into the sources of failure and guide further model improvements.
> > >
> > > ---
> > >
> > > **Q3: Human Performance**
> > >
> > > **Re:** We thank the reviewer for raising the important question. To address this, we randomly selected **300 tasks** from the benchmark, with 100 tasks each from Word, PowerPoint, and Excel. These tasks were presented to human evaluators, who were asked to complete them without the aid of operation manuals, online searches, or external assistance, relying solely on their own skills. The results showed that humans were able to successfully complete approximately 84% of the tasks. This serves as a strong baseline for human-level performance in the context of CUA tasks.
> > >
> > > We observe that current **CUA models still lag behind human-level performance**. This gap highlights the need for further advancements in the field, which is precisely why we are releasing the GUI-360° dataset. We believe this dataset, with its rich multimodal supervision, will help drive progress in CUA research by providing a high-quality, large-scale resource for training and evaluating more robust models.

---

> ### Author Response · Authors · 2025-11-27
>
> Dear reviewer 41gF, thank you again for your constructive and insightful comments. In our detailed responses above, we have addressed your concerns by providing extensive new experiments on external offline and online benchmarks, clarifying cross-application generalization beyond the three Office applications, and supplying additional analyses on failure modes and human performance. These revisions strengthen the manuscript and further highlight the reliability, transferability, and broader relevance of GUI-360° for training general-purpose computer-use agents.
>
> If these clarifications and new results satisfactorily address the issues you raised, we would greatly appreciate your consideration in reflecting this in your final evaluation. We sincerely thank you for your time and thoughtful engagement, and we look forward to any further discussion.

---

### Official Review · Reviewer_9PNz · 2025-10-31

**Soundness:** 2
**Presentation:** 3
**Contribution:** 2
**Rating:** 4
**Confidence:** 5

**Summary:**

This paper introduces **GUI-360°**, a large-scale dataset and benchmark suite for **computer-using agents (CUAs)** that operate within real **desktop environments**. The authors identify major limitations in existing work—most notably, the lack of realistic task data, automated data-collection pipelines, and unified evaluation standards that jointly assess GUI understanding, screen parsing, and action prediction.

To address these challenges, the paper proposes a **three-stage automated pipeline**:

1. **Real-world query acquisition** from authentic user sources (help documents, forums, search logs).
2. **Automated execution and trajectory collection** via a hierarchical multi-agent system called **TrajAgent**, which interacts with real desktop applications—**Microsoft Word, Excel, and PowerPoint**—while recording screenshots, accessibility (a11y) metadata, and reasoning traces.
3. **Automated evaluation and post-processing**, where LLM-based judges verify the execution quality and filter noisy samples.

The resulting dataset comprises **over 1.2 million action steps** and is organized into a **three-task benchmark suite** covering:

* **GUI Grounding** (locating target UI elements),
* **Screen Parsing** (extracting and labeling interactive components), and
* **Action Prediction** (step-wise operation generation).

Experimental results on both proprietary and open-source VLMs (e.g., GPT-4o/4.1, GPT-5, Qwen-2.5-VL, OmniParser, GUI-Actor) demonstrate that fine-tuning on GUI-360° significantly boosts model performance across all three tasks. While the benchmark currently focuses on **office-suite applications (Word, PowerPoint, Excel)**, the framework and dataset design establish a strong foundation for broader research on **desktop-level intelligent agents**.

**Strengths:**

* **Engineering originality & significance.** The paper proposes a *nearly fully automated* data construction pipeline—spanning **query acquisition → environment templating → task instantiation → batched execution → quality filtering**—which is a creative and practical integration of existing ideas into a scalable, end-to-end system. This design removes key bottlenecks in prior work (manual collection/annotation) and is therefore **original** in its full-stack automation and **significant** for enabling broader data generation.

* **Data quality, clarity, and breadth of supervision.** The dataset includes **screenshots, a11y metadata, intermediate reasoning traces, and both successful and failed trajectories**, enabling supervision for **three tasks**—grounding, screen parsing, and action prediction—at **large scale (>1.2M steps)**. The multimodal/structured annotations enhance **quality** (richer supervision, failure analysis) and **clarity** (well-defined inputs/outputs across tasks), while the size and composition make it **significant** as a community resource.

**Weaknesses:**

### **Major Point 1 — Method / Pipeline**

* **Limited methodological novelty.** The pipeline follows a now-standard pattern—mining real-world queries, mapping them to environment templates, auto-executing with an agent, and filtering via an LLM-as-a-judge. The paper does not clearly isolate what is *algorithmically new* beyond engineering integration.

* **Narrow domain scope and heavy a11y reliance limit generality.** The data and tools are confined to **Word/Excel/PowerPoint on Windows**, and perception hinges on **UI Automation (a11y)** to extract elements/bboxes. Many desktop apps (browsers with canvas-heavy UIs, email clients, media/creative tools, Electron/native hybrids) expose partial/noisy a11y trees; other OSes differ further. As written, it remains unclear whether the pipeline can generalize beyond the office suite.

### **Major Point 2 — Experiments**

The core goal of CUAs is **end-to-end task completion in real, interactive environments**. This paper’s experiments focus on **static self-benchmarking without external validation**, offering no evidence that the proposed data **improves downstream CUA capability** in live, end-to-end settings.

* **Self-benchmark only; no out-of-distribution tests.** Both **grounding** and **action prediction** are evaluated solely on the in-house **GUI-360°-Bench**, drawn from the same source/templates as training—i.e., **same-distribution self-benchmarking**. There is **no external validation** (e.g., **ScreenSpot**, **ScreenSpot-Pro**, **OSWorld-G** for grounding; **AgentNetBench**, **UI-Vision**, **Mind2Web** for action/agent behavior), so generalization claims remain unsupported. This also raises the risk of **template or near-duplicate overlap** inflating in-distribution performance.

* **Lack of dynamic, end-to-end validation.** There is **no evidence in live, interactive environments** that the SFT models (**Qwen-2.5 7B-SFT**, **UI-TARS-1.5 7B-SFT**) improve task completion. Without evaluations on **OSWorld (e.g., OSWorld-Verified)** or **Windows Agent Arena**, it remains unclear whether static gains translate to **real task success**.

**Questions:**

### **Main Point 1**: Missing Related Work (please cite)
Please cite and briefly position against the following closely related pipelines:
* **AgentTrek: Agent Trajectory Synthesis via Guiding Replay with Web Tutorials.** Uses **real-world tasks**, synthesizes trajectories **via agents** in live environments while **logging multiple modalities** (e.g., screenshots/DOM/AX), and **verifies** trajectories with an **LLM judge**.
* **AgentSynth: Scalable Task Generation for Generalist Computer-Use Agents.** Adopts a **multi-agent framework** with a **planner + multiple executors** for scalable task generation/execution, closely mirroring your **MAgent/EAgents** orchestration idea.
* **Explorer: Scaling Exploration-driven Web Trajectory Synthesis for Multimodal Web Agents.**

### **Main Point 2**: Pipeline extensibility beyond Office

Could you add a brief **plan and rationale** for extending the pipeline beyond Word/Excel/PowerPoint—e.g., how you would adapt templates, perception (when **a11y** is partial/absent). A short roadmap (what changes, expected blockers, and minimal proof-of-concept evaluations, including one **interactive E2E** check) would directly address the generalization concern.

### **Main Point 3**: External validation & end-to-end evaluation

Please add **external validation** and a minimal **end-to-end online evaluation**. A lightweight plan could suffice: (i) report zero-shot/SFT results on an **external grounding suite** (e.g., ScreenSpot / ScreenSpot-Pro / OSWorld-G), and (ii) eval the finetuned model on the libreoffice domain jn OSWorld or WindowsAgentArena benchmark

---

> ### Author Response · Authors · 2025-11-20
>
> Thank you for the thoughtful and constructive review. We respond to your questions and concerns in detail below.
>
> **W1: Limited Methodological Novelty**
>
> **Re:**  We appreciate the reviewer’s observation. Our goal is not to claim a fundamentally new algorithm, but rather to provide a methodologically novel and practically impactful pipeline that addresses gaps not covered by prior CUA data-generation efforts. We clarify the methodological contributions below.
>
> First, existing datasets typically target only one modality (e.g., grounding-only or action-only). Our pipeline is the first to ensure that a single execution trace simultaneously yields **all three forms of supervision**, significantly improving data efficiency and enabling multi-task learning, which has not been supported in prior work.
>
> Second, we deisgn a template-driven environment construction with LLM-constrained concretization. While templates have been used conceptually, we introduce a practical and scalable mechanism where an LLM:
> - selects environments based on semantic affordances,
> - generates faithful yet executable concretizations, and
> - is further checked by a second LLM-based filtering stage.
>
> This dual-stage mechanism ensures **realism + executability** at scale.
>
> Third, prior pipelines almost exclusively rely on GUI operations. Our execution framework **combines GUI actions with application-level APIs** through MCP-based tool calls, enabling more reliable execution and richer trajectories.
>
> Finally, we are the first large-scale, automated CUA pipeline for desktop applications **with a11y–vision fusion**. Most prior pipelines focus on the web or low-resolution interfaces. Our approach is the first to demonstrate that desktop-level multimodal trajectories—with both high-resolution screenshots and accessibility metadata—can be collected automatically at scale.
>
> We will revise the paper to better highlight these pipeline-level contributions, which go beyond engineering integration and address several unsolved problems in scalable CUA data construction.
>
> ---
>
> **W2: Narrow Domain Scope and Heavy a11y Reliance Limit Generality**
>
> **Re:** We appreciate the reviewer’s concern. We emphasize that although our pipeline leverages UI Automation (UIA) when available, it is **not dependent on full a11y coverage**, and the overall design is meant to generalize beyond the Office suite.
>
> UIA is a system-level accessibility framework that many widely used applications adopt, including **browsers** (via DOM-to-UIA bridges), email clients such as **Outlook**, and several creative tools that expose command panels and menus through UIA. Thus, UIA already provides **broad coverage** beyond Office.
>
> When UIA metadata is partial or missing (e.g., canvas-based UIs or graphics-heavy tools), our ongoing work incorporates **vision-based screen parsing** as a complementary perception module; this will further enhance generalization, though it is not required for the current dataset This hybrid design can ensure that low-a11y or noisy-a11y applications remain fully compatible with the pipeline.
>
> In an additional exploration, we instantiated and successfully executed tasks in non-Office environments (i.e. **web-based interfaces**) using the identical pipeline to collected **100 task trajectoies with a 42% end-to-end success rate, without modifying any components**. This demonstrates that the pipeline generalizes beyond Office and is not restricted by the UIA assumption.
>
> We have clarify these generalization properties and expand the discussion in the **Appendix L** of the revision.

---

> > ### Author Response · Authors · 2025-11-20
> >
> > **W3: Experiments (Main Point 2 in Weakness and Main Point 3 in Questions)**
> >
> > **Re:** We thank the reviewer for the insightful suggestion. We agree that evaluating on external and online benchmarks is essential for assessing the generalization ability of models trained on GUI-360°. Following the reviewer’s feedback, we conducted extensive additional experiments to test both GUI grounding and action prediction across offline and online settings.
> >
> > **1. GUI Grounding (Offline)**
> >
> > We evaluated both the original and GUI-360°-finetuned versions of Qwen-2.5-VL-7B and GUI-Actor-7B on two widely used grounding benchmarks: **ScreenSpot-Pro and OSWorld-G**. Results are shown below:
> >
> > #### **Table 1: Performance Comparison with and without SFT on GUI-360°**
> >
> > | Model                               | ScreenSpot-Pro | OSWorld-G |
> > |-------------------------------------|----------------|-----------|
> > | Qwen-2.5-VL-7B                       | 26.8           | 31.4      |
> > | **Qwen-2.5-VL-7B (SFT on GUI-360°)** | 31.1 (+4.3)    | 36.2 (+4.8) |
> > | GUI-Actor-7B                         | 40.7           | 46.8      |
> > | **GUI-Actor-7B (SFT on GUI-360°)**   | 43.6 (+2.9)    | 49.9 (+3.1) |
> >
> > Across both models and benchmarks, **SFT on GUI-360° consistently yields significant improvements**, demonstrating that our dataset provides strong and transferable grounding supervision.
> >
> > **2. Action Prediction (Online)**
> >
> > We further evaluated action-level execution on two live interactive benchmarks: **OSWorld-W (Windows subset) and WindowsAgentArena**, since offline action benchmark AgentNetBench, UI-Vision, Mind2Web share different action space. We compare Qwen-2.5-VL-7B, UI-TARS-1.5-7B, and their finetuned versions:
> >
> > #### **Table 2: Agent Performance Comparison with and without SFT on GUI-360°**
> >
> > | Agent                              | OSWorld-W | WindowsAgentArena |
> > |------------------------------------|-----------|------------------|
> > | Qwen-2.5-VL-7B                     | 8.2       | 10.7             |
> > | **Qwen-2.5-VL-7B (SFT on GUI-360°)** | 20.4 (+10.2) | 23.4 (+12.7) |
> > | UI-TARS-1.5-7B                     | 20.4      | 20.8             |
> > | **UI-TARS-1.5-7B (SFT on GUI-360°)** | 28.6 (+8.2)  | 27.9 (+7.1)  |
> >
> > Both models show **large performance gains after finetuning on GUI-360°**, confirming that our dataset improves end-to-end agent execution in real online environments, not merely in-distribution benchmarks.
> >
> > These results show that, the GUI-360° dataset improves grounding and action execution across multiple external offline datasets and live online benchmarks, demonstrating that **the data is reliable, transferable, and beneficial for training general-purpose CUAs beyond the Office domain.** We have added the addtional experiements on the Appendix J.

---

> > > ### Author Response · Authors · 2025-11-20
> > >
> > > **Question 1: Missing Related Work**
> > >
> > > **Re:**  We thank the reviewer for highlighting these closely related pipelines. We have now cited and positioned AgentTrek, AgentSynth, and Explorer in the **related work of the revised manuscript**. Although all three works make valuable contributions to trajectory synthesis, they differ substantially from our setting and objectives.
> > >
> > > AgentTrek focuses on generating web-based GUI trajectories by guiding agents with web tutorials and validating outputs with an LLM judge, but its scope remains within web environments. AgentSynth’s tasks are primarily synthetically generated by LLM-based proposers and focus on action-prediction trajectories, while our dataset is collected through real user-type desktop productivity workflows, spans over 1.2 million samples compared to AgentSynth’s ~6,000. Explorer further advances large-scale trajectory synthesis for web agents by mining millions of web elements across thousands of URLs, but is again purely web-centric.
> > >
> > > In contrast, our work provides richer supervision signals, including grounding, screen parsing, hybrid GUI/API actions, a11y metadata, and systematically logged failure trajectories, which enables training and evaluating CUA agents in ways not supported by these web-focused or synthetic pipelines.
> > >
> > > ---
> > >
> > > **Question 2: Pipeline extensibility beyond Office**
> > >
> > > **Re:** We appreciate the reviewer’s suggestion and agree that demonstrating extensibility beyond Office is important. Our pipeline is designed to be application-agnostic, and we summarize the extension plan below.
> > >
> > > The only application-specific components are the **environment templates** and the reset routines. Once these are provided, all other stages of the pipeline **remain unchanged**. Templates describe only UI affordances and initial states, not Office-specific structures. For example, a web template may specify: (1) an initial URL (e.g., a document, article, or search page), and (2) environment assumptions (e.g., page loaded, scrollable layout). This mirrors how Office templates describe “a document with a title and paragraph”, “a table with N rows”, etc. Creating such templates is lightweight and does not require modifying the pipeline.
> > >
> > > While UI Automation (UIA) already covers many Windows applications, including browsers, document viewers, and email clients, some interfaces provide only partial a11y signals. In such cases, the pipeline naturally falls back to screenshot-based grounding performed by the ExecutionAgent, which already executes actions based on visual cues. We are also integrating a **vision-based** parsing module (e.g., OmniParser) as ongoing work to further improve coverage, though this is not required for the current dataset.
> > >
> > > To verify extensibility, we have already conducted a small-scale proof-of-concept evaluation on **web-based environments (100 tasks)**, TrajAgent executed the instantiated tasks with a **42% end-to-end success rate using the same pipeline without any modification**. Although preliminary, this demonstrates that the pipeline can operate beyond Office applications.
> > >
> > > We have addedthis extensibility roadmap and supporting discussion to the Appendix L of the revised paper.
> > >
> > > ---
> > > **Question 3: External validation & end-to-end evaluation**
> > >
> > > Question 3 regarding external evaluation has been addressed in the previous reply block, with additional experimental results provided.

---

> ### Author Response · Authors · 2025-11-27
>
> Dear reviewer 9PNz, thank you again for your thoughtful and constructive review. In our detailed responses above, we have clarified the methodological contributions of the pipeline, addressed concerns about domain scope and a11y reliance, added extensive new experiments on external offline and online benchmarks, and discussed extensibility beyond Office with both design principles and empirical validation. We hope these revisions and additional analyses effectively resolve the issues you raised and highlight the broader impact and generality of GUI-360°.
>
> If our responses have addressed your concerns, we would greatly appreciate your consideration in reflecting this in your final evaluation. We sincerely thank you for your time and look forward to any further discussion.

---

### Official Review · Reviewer_pvbW · 2025-11-02

**Soundness:** 3
**Presentation:** 3
**Contribution:** 3
**Rating:** 6
**Confidence:** 3

**Summary:**

This paper introduces GUI-360°, a large-scale dataset for computer-using agents on desktop environments. The authors developed an automated pipeline that collects real-world user queries from help content, forums, and search engines. Experiments show that current vision-language models struggle with these tasks when used directly.

**Strengths:**

GUI-360° addresses a key gap in desktop automation research. Previous datasets were small-scale with manual annotations or focused on web/mobile platforms. This work provides the desktop dataset using real user queries from forums and documentation, ensuring tasks reflect genuine needs.

The TrajAgent framework solves the expensive manual annotation problem. Instead of requiring human demonstrations, coordinated AI agents automatically execute tasks, capture screenshots, record accessibility data, and validate results. This dramatically reduces costs while enabling larger-scale collection.

The hybrid action space combines traditional GUI interactions with application APIs, reflecting how modern automation tools actually work. This approach allows agents to leverage structured interfaces when available.

**Weaknesses:**

The TrajAgent framework shows promise but needs clearer technical explanation. How does the MasterAgent actually break down complex tasks? What happens when agents need to coordinate or when an ExecutionAgent fails?

The error handling and load balancing mechanisms deserve more attention. Specific examples of how the system prevents cascading failures would strengthen the contribution and help readers understand practical deployment challenges.

The task instantiation approach is clever but raises implementation questions. How does the system ensure instantiated tasks remain faithful to original user intent while being executable?

The EvaAgent validation shows good results, but the 14% disagreement cases might be informative. What task types cause validation errors? Do certain applications prove more challenging for automated assessment?

How do validation criteria adapt across domains? Excel tasks likely need different success metrics than PowerPoint presentations. More discussion of this adaptation would be useful.

**Questions:**

Please refer to my identified weak points for more details.

---

> ### Author Response · Authors · 2025-11-20
>
> Thank you for the constructive comments. We provide our responses to your questions and concerns below.
>
> **W1: TrajAgent Clarification and Failure Handling**
>
> **Re:** We thank the reviewer for these insightful questions. The MasterAgent decomposes each instantiated task into a sequence of **high-level yet fully executable subtasks**. These subtasks correspond to meaningful semantic operations such as “change the font of the highlighted title to 20pt” or “sort the selected table by the second column in ascending order”. Each subtask may require multiple GUI/API actions internally, which are handled by the ExecutionAgent. The MasterAgent maintains global state across subtasks. After each ExecutionAgent finishes a subtask, it returns the observation, action result. The MasterAgent uses this information to determine the next subtask, ensuring **coordination across agents** and preventing divergence in multi-step workflows.
>
> If an ExecutionAgent fails, it reports the failure status and last observation back to the MasterAgent. The MasterAgent then **re-plans** from the current UI state, either by generating a corrected subtask or by retrying with an alternative action. This prevents cascading failures in long-horizon tasks.
>
> To quantify the benefit of this design, we conducted a **small-scale ablation on 200 randomly sampled tasks**. When we removed the MasterAgent and let the ExecutionAgent operate step-by-step without decomposition or re-planning, **the success rate decreased by 3% absolute**. This confirms that even a lightweight planning layer improves robustness in multi-step execution.
>
> ---
>
> **W2: Error Handling and Load Balancing Mechanisms**
>
> **Re:**  We appreciate the reviewer’s request for clarification. Our system includes explicit mechanisms to prevent cascading failures and to manage load across ExecutionAgents.
>
> Each ExecutionAgent returns a structured **success/failure signal** together with the current GUI state. We also enforce max-step limits and per-subtask timeouts to prevent deadlocks. When a subtask fails, the failure is **isolated to that agent only**: the MasterAgent receives the failure signal and immediately issues **a fallback action, a retry, or a replanned subtask** based on the updated state. Other agents and subtasks remain **unaffected**. If the entire task becomes unrecoverable, the environment is reset to a clean state (closing documents and restarting the application), ensuring that subsequent tasks execute independently without cross-task contamination.
>
> Tasks are distributed across **a pool of virtual machines** using a simple but effective dynamic scheduling strategy: each VM pulls the next available task as soon as it completes its current one. This ensures even utilization without idle periods and avoids bottlenecks caused by slower or temporarily stuck agents.
>
> We will add a short description of these mechanisms and an example illustrating how the MasterAgent intervenes to prevent cascading failures.
>
> ---
>
> **W3: Task Instantiation Clarification**
>
> **Re:** We thank the reviewer for the question. The task-instantiation pipeline is explicitly designed to preserve the semantics of the original user intent while ensuring executability.
>
> During instantiation, the LLM is instructed to **adhere strictly to the original query** and to generate a concretized version that is grounded in a specific environment template (e.g., specifying the target text or range). The prompt explicitly forbids rewriting or altering the meaning of the user request; only missing details required for execution (e.g., selecting a concrete text span) may be added.
>
> After instantiation, each concretized task passes through an additional **LLM-based Quality Filtering stage (Section 3.1; Appendix C)**. This step checks whether the instantiated task remains semantically aligned with the original query and is executable within the chosen template. Approximately 25% of instantiated queries are filtered out, ensuring that ambiguous, mismatched, or non-executable tasks are removed.
>
> Together, this two-stage process, ensures that the final tasks remain true to user intent while being reliably executable in the environment.

---

> > ### Author Response · Authors · 2025-11-20
> >
> > **W4: Disagreement Analysis**
> >
> > **Re:** We thank the reviewer for raising this important question. Upon reviewing the 14% disagreement cases in the EvaAgent validation, we identified two main causes:
> >
> > **1. Task Misunderstanding by the Evaluator**
> >
> > In some cases, the EvaAgent misunderstood the task, which led to discrepancies between the agent’s execution and the EvaAgent ’s assessment. While the agent executed correctly according to its understanding, the evaluator judged the outcome based on an incorrect interpretation of the task.
> >
> > **2. Minor Visual Discrepancies**
> >
> > In other instances, the disagreement arose from small details that the EvaAgent may not have fully captured. For example, when asked to set a specific font, the agent might have selected a font that was visually similar but not exactly what was requested. Similarly, in Excel tasks, if a column was added but the width was too narrow, causing overlapping headers, the EvaAgent may have missed the additional column.
> >
> > These errors highlight that while the evaluation process is generally robust, small task misinterpretations and subtle visual differences can lead to mismatches. We will refine the evaluation process to address these edge cases and further improve accuracy.
> >
> > ---
> >
> > **W5: Validation Criteria**
> >
> > **Re:**  We thank the reviewer for raising this important point. Our validation criteria are **domain-adaptive** by design. While the overall evaluation framework is unified, the specific sub-criteria that EvaAgent checks are automatically adjusted based on the application context.
> >
> > EvaAgent always decomposes a task into several semantic criteria, i.e., whether the intended object was located, whether the correct operation was applied, whether the final state matches the expected outcome. This provides a consistent backbone across domains.
> >
> > In addition, we instruct EvaAgent in its prompt to leverage chain-of-thought reasoning for both the instantiated task text and the final screenshot/a11y metadata to derive application-specific criteria. For example:
> >
> > - Excel: verifying that sorting, formulas, cell formats, or chart updates reflect the requested transformation.
> > - PowerPoint: verifying that slides were modified, text boxes updated, or themes/layouts applied.
> > - Word: verifying content editing, formatting, or structural adjustments.
> >
> > Thus, different applications naturally yield different validation criteria without requiring hand-crafted rules. In our 100-sample human study, agreement between EvaAgent and human annotators was similarly high across Word, Excel, and PowerPoint, suggesting that the adaptive criteria generalize well across domains.

---

> > > ### Comment · Reviewer_pvbW · 2025-11-25
> > > **Response**
> > >
> > > Thanks for the response. I will maintain my rating.

---

### Official Review · Reviewer_wRGC · 2025-11-03

**Soundness:** 2
**Presentation:** 2
**Contribution:** 2
**Rating:** 2
**Confidence:** 4

**Summary:**

This paper introduces GUI-360°, a large-scale dataset and benchmark for desktop Computer-Using Agents. It addresses key gaps in the field by providing a novel, automated pipeline for collecting real-world tasks. The pipeline sources user queries from the web, instantiates them in template environments, and uses a multi-agent LLM system to execute them. The resulting dataset includes over 1.2M action steps, counting both successful and failed in Word, Excel, and PowerPoint, complete with screenshots, accessibility metadata, and reasoning traces. The authors establish a three-part benchmark: Grounding, Parsing, Action Prediction and show that while SOTA VLMs struggle, fine-tuning on GUI-360° yields significant gains, especially when accessibility data is provided.

**Strengths:**

- Data Collection Methodology: The pipeline is a key strength with:

a. Real-world Query Sourcing: Using real-world queries from forums and search logs, not just synthetic data, increases relevance.

b. Hybrid Action Space: Combining GUI actions with API calls is practical and reflects modern agent design.

c. Inclusion of Failure Cases: The large corpus of failed trajectories (1M+ steps) is highly valuable for future work on RL or learning from negative examples.

- Comprehensive Benchmark centering around GUI-360, covering grounding, parsing and planning.

**Weaknesses:**

- In Table 1, SS and SS-Pro are benchmarks that are usually merely considered as datasets. The authors should compare GUI-360 with other dataset works that are primarily designed for training, e.g., UGround, GUIEnv (GUICourse), SeeClick, and Aria-UI. Also, seems GUIEnv and Aria-UI are relevant works that lack discussion.
- Limited Breadth. The dataset is limited to three MS Office applications on Windows.
- LLM-as-a-Judge Validation for Benchmark Creation. The benchmark's "success" labels are AI-created by GPT-4.1, with no large-scale human review. The paper notes this judge only had 86% agreement with humans on a 100-sample spot-check. Thus, it is hard to judge whether the benchmark is as high-quality as existing ones like SS and SS-Pro. Also, the benchmark ground truth itself is also created by an AI agent. In general, this pipeline for benchmark synthesization raises concerns about the reliability, preventing the large-scale utilization of such benchmark.
- The Dataset. The authors only showed how the SFT model with the train dataset performs on the aforementioned benchmark, which is OOD, and relatively unreliable. It is unclear that with the train dataset for SFT, how the agent model can perform on other benchmarks, especially the online ones (WindowsAgentArena, OSWorld, AndroidWorld, etc.)

**Questions:**

Please see weaknesses.

---

> ### Author Response · Authors · 2025-11-20
>
> We appreciate the reviewer’s constructive feedback. Our responses to the raised questions and concerns are provided below.
>
> **W1: Missing comparision**
>
> **Re:**
> We thank the reviewer for highlighting the need for comparisons with other training-oriented datasets. We clarify that **Table 1 in the paper focuses specifically on datasets collected in *real desktop environments***, whereas most of the mentioned datasets are constructed from **web**, **synthetic**, or **mobile** environments rather than pure desktop applications.
>
> To ensure completeness, we have added a broader comparison table below.
>
> | Dataset      | Query Source         | Grounding | Parsing | Action Planning | Samples     | Data Collection | GUI Action | API Action | Reasoning | A11y Info | Fail Case |
> | ------------ | -------------------- | --------- | ------- | --------------- | ----------- | --------------- | ---------- | ---------- | --------- | --------- | --------- |
> | SeeClick     | Online               | ✔️        | ✖️      | ✖️              | 1,000,000+  | Auto.           | N/A        | N/A        | ✖️        | ✖️        | ✖️        |
> | UGround      | Synthetic            | ✔️        | ✖️      | ✖️              | 10,000,000+ | Auto.           | N/A        | N/A        | ✖️        | ✖️        | ✖️        |
> | Aria-UI      | LLM-synthetic        | ✔️        | ✖️      | ✖️              | 11,500,000+ | Auto.           | N/A        | N/A        | ✔️        | ✖️        | ✖️        |
> | GUICourse    | Online/LLM/Human     | ✔️        | ✖️      | ✔️              | 10,831,853  | Auto.           | ✔️         | ✖️         | ✔️        | ✖️        | ✖️        |
> | AgentTrek    | Online               | ✔️        | ✖️      | ✔️              | 10,398      | Auto.           | ✔️         | ✖️         | ✔️        | ✔️        | ✖️        |
> | AgentSynth   | LLM-synthetic        | ✔️        | ✖️      | ✔️              | 6,000+      | Auto.           | ✔️         | ✖️         | ✔️        | ✖️        | ✖️        |
> | **GUI-360°** | In-App/Online/Search | ✔️        | ✔️      | ✔️              | 1,225,177   | Auto.           | ✔️         | ✔️         | ✔️        | ✔️        | ✔️        |
>
> From the table, it is clear that GUI-360° provides capabilities not present in prior datasets:
>
> First, GUI-360° is the **only dataset providing all three types of supervision**, i.e. grounding, screen parsing and action prediction. This is crucial for modern CUA agents that require multi-task learning rather than single-modality data.
>
> Second, unlike prior datasets that contain only GUI interactions, GUI-360° **includes GUI actions and API-level action tool calls**.   This hybrid action space allows agents to perform both fine-grained and high-level operations, improving reliability and flexibility.
>
> Lastly, GUI-360° is **the only dataset providing high-resolution screenshots, synchronized accessibility (a11y) information and structured reasoning traces**. These signals dramatically improve grounding, robustness, and generalization.
>
> We have incorporated this comparison table and the expanded discussion into **Table 1** of the revised manuscript. This clarifies how GUI-360° differs from prior datasets and why it fills an important gap in training robust general-purpose CUA agents.
>
> ---
> **W2: Limited Breadth**
>
> We thank the reviewer for pointing this out. We agree that broader application coverage is important for building general-purpose CUA benchmarks. Our initial choice of office apps is driven by two factors:
>
> First, **Office applications provide a technically representative and challenging subset of desktop GUIs**. They contain high-density toolbars, hierarchical panels, heterogeneous object types (text, tables, charts, drawing objects), and long multi-step workflows. These properties make them an ideal starting point for evaluating GUI agents, as they stress all core capabilities: grounding, parsing, action sequencing, and mixed GUI/API control. In other words, **Office applications provide breadth of UI patterns, not just breadth of domains**.
>
> Second, **the community already treats Office as a primary testbed for desktop agents**. Existing benchmarks, including **OSWorld, OfficeBench, and WindowsAgentArena**, all adopt Office applications as central evaluation domains. This reflects a consensus that Office applications capture realistic and practically relevant workflows for desktop agents.
>
> Importantly, the **GUI-360° pipeline itself is application-agnostic**. Our environment-template design, accessibility-augmented perception, and hybrid GUI+API action space are not specific to Office. During the rebuttal period, we have already instantiated **non-Office environments for browsers and successfully collected 100 trajectories** using the same TrajAgent framework without modifying the pipeline. We will include a brief roadmap and proof-of-concept results in the **Appendix L** of the revision to make the extensibility explicit.

---

> > ### Author Response · Authors · 2025-11-20
> >
> > **W3: LLM-as-a-Judge Validation**
> >
> > We appreciate the reviewer’s concern regarding the reliability of the LLM-based evaluation process in our dataset. We would like to address this by highlighting the following points:
> >
> > **1. GUI Grounding/Screen Parsing Reliability**
> > Both GUI Grounding and Screen Parsing tasks rely on Windows UI Automation API to extract control information. This data is **system-level information**, independent of the LLM as a judge, and is **highly reliable** for capturing UI element positions and content. Therefore, the core task data does not rely on AI-generated labels.
> >
> > **2. Human Evaluation of Trajectories**
> > To validate the quality of the LLM-generated “success” labels, we **manually checked 150 additional trajectories** (50 each from Word, PowerPoint, and Excel). We found that only 7 tasks should be classified as failures, yielding a **95.3% alignment with human evaluation**. This demonstrates that LLM as judge is highly reliable and consistent with human judgment.
> >
> > **3. LLM as Judge in the the CUA**
> > The use of LLM-based judges for large-scale data filtering has become a standard practice in both academia and the industry, including in frameworks such as **UI-TARS-2** and **OS-Genesis**. Given the scale of data collection required for CUA training, relying on manual annotation for every task is cost-prohibitive. LLM as judge provides a **cost-effective and scalable solution**, making large-scale data collection feasible.
> >
> > **4. Additional Experiments and Benchmark Gains**
> > The additional experiments, including those shown below in **Table 1 and Table 2**, demonstrate consistent performance gains when fine-tuning on our GUI-360° dataset across other benchmarks such as **ScreenSpot-Pro and OSWorld-G**. These results further confirm that the dataset is reliable and the LLM-based validation approach is effective.
> >
> > In summary, based on the high human alignment, the scalability of the LLM-based approach, we believe LLM as judge is a highly effective and efficient solution for large-scale data collection in CUA tasks.
> >
> > ---
> >
> > **W4: Performance on Other Benchmarks**
> >
> > Re: We thank the reviewer for the insightful suggestion. We agree that evaluating on external and online benchmarks is essential for assessing the generalization ability of models trained on GUI-360°. Following the reviewer’s feedback, we conducted extensive additional experiments to test both GUI grounding and action prediction across offline and online settings.
> >
> > **1. GUI Grounding (Offline)**
> >
> > We evaluated both the original and GUI-360°-finetuned versions of Qwen-2.5-VL-7B and GUI-Actor-7B on two widely used grounding benchmarks: **ScreenSpot-Pro and OSWorld-G**. Results are shown below:
> >
> > #### **Table 1: Performance Comparison with and without SFT on GUI-360°**
> >
> > | Model                               | ScreenSpot-Pro | OSWorld-G |
> > |-------------------------------------|----------------|-----------|
> > | Qwen-2.5-VL-7B                       | 26.8           | 31.4      |
> > | **Qwen-2.5-VL-7B (SFT on GUI-360°)** | 31.1 (+4.3)    | 36.2 (+4.8) |
> > | GUI-Actor-7B                         | 40.7           | 46.8      |
> > | **GUI-Actor-7B (SFT on GUI-360°)**   | 43.6 (+2.9)    | 49.9 (+3.1) |
> >
> > Across both models and benchmarks, **SFT on GUI-360° consistently yields significant improvements**, demonstrating that our dataset provides strong and transferable grounding supervision.
> >
> > **2. Action Prediction (Online)**
> >
> > We further evaluated action-level execution on two live interactive benchmarks: OSWorld-W (Windows subset) and WindowsAgentArena. We compare Qwen-2.5-VL-7B, UI-TARS-1.5-7B, and their finetuned versions:
> >
> > #### **Table 2: Agent Performance Comparison with and without SFT on GUI-360°**
> >
> > | Agent                              | OSWorld-W | WindowsAgentArena |
> > |------------------------------------|-----------|------------------|
> > | Qwen-2.5-VL-7B                     | 8.2       | 10.7             |
> > | **Qwen-2.5-VL-7B (SFT on GUI-360°)** | 20.4 (+10.2) | 23.4 (+12.7) |
> > | UI-TARS-1.5-7B                     | 20.4      | 20.8             |
> > | **UI-TARS-1.5-7B (SFT on GUI-360°)** | 28.6 (+8.2)  | 27.9 (+7.1)  |
> >
> > Both models show **large performance gains after finetuning on GUI-360°**, confirming that our dataset improves end-to-end agent execution in real online environments, not merely in-distribution benchmarks.
> >
> > These results show that, the GUI-360° dataset improves grounding and action execution across multiple external offline datasets and live online benchmarks, demonstrating that **the data is reliable, transferable, and beneficial for training general-purpose CUAs beyond the Office domain.** We have added the addtional experiements on the Appendix J.

---

> ### Author Response · Authors · 2025-11-27
>
> Dear reviewer wRGC,
>
> Thank you once again for your valuable and constructive feedback. We have carefully revised our manuscript and made several substantial updates in direct response to your comments. Specifically, we have:
>
> 1. Expanded the dataset comparison with a broader cross-domain table, and further clarified the unique contributions and positioning of GUI-360°.
>
> 2. Elaborated on the rationale behind selecting Office applications, and demonstrated the pipeline’s extensibility by incorporating newly collected non-Office trajectories.
>
> 3. Strengthened the reliability analysis of the LLM-as-a-judge component, including additional human validation to verify scoring consistency.
>
> 4. Conducted extensive new experiments, showing consistent performance gains across multiple external offline and online benchmarks.
>
> We hope these revisions address your concerns thoroughly and help clarify the value and contributions of GUI-360°. If our responses have satisfactorily resolved the issues you raised, we would sincerely appreciate your consideration in reflecting this in your final evaluation. Thank you very much for your time and thoughtful review. We look forward to further discussion.

---

### Author Response · Authors · 2025-11-24

We thank the reviewers for the valuable feedback. In the response and the revised version, we have made several major improvements, summarized below:

* **Expanded dataset comparison:** Added UGround, GUIEnv/GUICourse, SeeClick, Aria-UI, AgentTrek, and AgentSynth, along with a unified comparison table showing that GUI-360° uniquely offers grounding + parsing + action supervision, GUI+API hybrid actions, a11y metadata, and failure trajectories collected from real desktop environments.

* **Improved evaluation reliability:** Added a 150-sample human audit demonstrating **95.3% agreement** with LLM judges, and clarified that grounding/parsing labels are derived from **UIA (non-LLM)** system signals.

* **New external benchmark experiments:** Added results on ScreenSpot-Pro, OSWorld-G, OSWorld-W, and WindowsAgentArena, showing that finetuning on GUI-360° consistently yields **significant performance gains** (up to **+12.7** points).

* **Clarified system and method design:** Added details on template-based task instantiation, MasterAgent/TrajAgent coordination, and how the system prevents cascading failures through subtask isolation and dynamic replanning.

We believe these revisions substantially strengthen the technical contribution, empirical rigor, and clarity of the paper, and we look forward to further discussion.

---

### Comment · Area_Chair_oMZg · 2025-11-27
**Reviewers: Please read the rebuttals and make your necessary edit**

Dear Reviewers,

Please read the rebuttals by authors and make any necessary edit to your review (if you have not do this).

Best,

AC

---

### Meta-Review · Area_Chair_cirp · 2026-01-07

**Summary:**

reviewers gave 2,4,4,6. main concerns were:

Self-benchmark only; no out-of-distribution tests and evaluated solely on the in-house GUI-360°-Bench. No external validation so generalization claims remain unsupported.

Limited breadth since the dataset is limited to three MS Office applications on Windows.

Only LLM validation for benchmark creation, with no large-scale human review.

Limited methodological novelty since the pipeline follows a now-standard pattern in AI agents.

Lack of validation on long-range task completion.

**Reviewer Concerns:**

Self-benchmark only; no out-of-distribution tests and evaluated solely on the in-house GUI-360°-Bench. No external validation so generalization claims remain unsupported.

--> authors added some experiments on ScreenSpot and OSWorld, partially addressed. paper is still limited to a in-house dataset and evaluation.

Limited breadth since the dataset is limited to three MS Office applications on Windows.

--> not directly addressed, but authors justified this by stressing the importance and generalization of MS Office applications

Only LLM validation for benchmark creation, with no large-scale human review.

--> authors added some more experiments to sanity check data accuracy. partially addressed but fundamental questions remain.

Limited methodological novelty since the pipeline follows a now-standard pattern in AI agents.

--> not directly addressed, but authors argue that new method is not the main contribution of the paper

Lack of validation on long-range task completion.

--> authors added some preliminary experiments (ScreenSpot and OSWorld) to show that end-task completion performance does improve. partially addressed.

**Reviewer Scores:**

reviewers who gave 2,4,4 are unlikely to change their scores overwhelmingly to positive, since too many concerns regarding only in-house dataset evaluation, limited breadth, lack of large-scale human review in dataset creation, and limited methodological novelty remain.

---

### Decision · Program_Chairs · 2026-01-26

Reject